# Inverted signaling by bacterial chemotaxis receptors

Shuangyu Bi[1], Fan Jin[1] & Victor Sourjik[1]

Microorganisms use transmembrane sensory receptors to perceive a wide range of environmental factors. It is unclear how rapidly the sensory properties of these receptors can be modified when microorganisms adapt to novel environments. Here, we demonstrate experimentally that the response of an *Escherichia coli* chemotaxis receptor to its chemical ligands can be easily inverted by mutations at several sites along receptor sequence. We also perform molecular dynamics simulations to shed light on the mechanism of the transmembrane signaling by *E. coli* chemoreceptors. Finally, we use receptors with inverted signaling to map determinants that enable the same receptor to sense multiple environmental factors, including metal ions, aromatic compounds, osmotic pressure, and salt ions. Our findings demonstrate high plasticity of signaling and provide further insights into the mechanisms of stimulus sensing and processing by bacterial chemoreceptors.

[1] Max Planck Institute for Terrestrial Microbiology & LOEWE Center for Synthetic Microbiology (SYNMIKRO), Marburg 35043, Germany. These authors contributed equally: Shuangyu Bi, Fan Jin. Correspondence and requests for materials should be addressed to V.S. (email: victor.sourjik@synmikro.mpi-marburg.mpg.de)

Transmembrane receptors are ubiquitously used by prokaryotes to monitor their environment. Elucidating how receptors sense environmental stimuli is thus essential for understanding bacterial adaptation to various environmental niches. Important models for investigation of sensory mechanisms are chemotaxis receptors, which control cell motility to enable prokaryotes to accumulate towards favorable conditions[1,2]. This control is exerted by regulation of the autophosphorylation of the receptor-associated kinase CheA, which subsequently donates the phosphoryl group to the CheY response regulator that controls rotation of flagellar motor. In *Escherichia coli*, the best-studied model for chemotactic signaling, positive chemotactic stimuli (attractants) inhibit the activity of CheA, whereas negative stimuli (repellents) stimulate it. The chemotaxis signaling pathway in *E. coli* further includes the phosphatase CheZ that dephosphorylates CheY and an adaptation system that consists of the methyltransferase CheR and the methylesterase CheB, which modify receptors on several specific glutamyl residues.

Chemotaxis receptors have a modular structure (Fig. 1a) that is typically conserved throughout bacteria and archaea[3]. Variation in the sequence and structure of periplasmic sensory domains enables chemoreceptors to accommodate many different types of ligands[3,4]. In *E. coli* chemoreceptors, sensory domains are dimers of pseudo-four-helix bundles (α1–α4 and α1′–α4′)[5], whereby α1/α1′ and α4/α4′ extend into membrane as transmembrane helices TM1/TM1′ and TM2/TM2′, respectively. Ligand binding has been shown to promote a 1–2 Å inward piston displacement of the α4-TM2 helix relative to the α1-TM1 helix[2,6,7]. Signals are further transduced by the HAMP homodimer domain, which is a four-helix, parallel coiled coil composed of two amphipathic

**Fig. 1** Effects of mutations in TM2 on Tar response to MeAsp. **a** Schematic representation of *E. coli* Tar receptor. **b** Sequence alignment for the wild-type and mutant Tar. Different colors indicate receptors that mediate attractant, repellent, or no responses to MeAsp, respectively. **c**, **d** FRET responses of *E. coli* CheR+CheB+ strain VS181, expressing the wild-type Tar (**c**) or Tar^TM2+2I (**d**) as a sole receptor, as well as the FRET pair, to a stepwise addition (down arrow) and subsequent removal (up arrow) of the indicated concentrations of MeAsp. The corresponding amplitudes of initial FRET responses were plotted as a function of step changes in concentration of MeAsp. The data were normalized to the saturated response. Error bars indicate standard deviation of three independent biological replicates; note that for most data points error bars are smaller than the symbol size. **e**, **f** Microfluidic assay of the chemotactic response of receptorless strain UU1250 expressing wild-type Tar (**e**) and Tar^TM2+2I (**f**) to MeAsp. Cell distribution in the observation channel was recorded when the source is 10 mM MeAsp or without MeAsp (scale bar, 100 μm). The *x*-component (black arrow) indicates the direction up the concentration gradient of MeAsp. Relative cell density (fluorescence intensity) in the observation channel for the wild-type Tar (**e**) or Tar^TM2+2I (**f**) responding to MeAsp (closed squares) or without MeAsp (open squares) were plotted over time. The cell density in the observation channel before MeAsp stimulating (*t* = 0) was normalized to be one. Error bars indicate standard deviation of three independent biological replicates

helices AS1/AS1′ and AS2/AS2′[8,9]. The input and output signaling of the HAMP domain plays a central role in transmission of signals[10,11]. A junction between TM2 and AS1 of the HAMP domain is formed by a short control cable that extends from TM2 and has helical character[12,13]. Changes in the helicity of this control cable were proposed to be important in transmitting the piston motion of TM2 and modulating the conformation of the HAMP domain[12–16]. Subsequently, signals are transduced through the methylation helix (MH) bundle, the flexible region, and towards the protein contact region interacting with CheA and the adaptor protein CheW, with all of these regions organized as one anti-parallel four-helix bundle[17,18]. Structural dynamics of these cytoplasmic regions are thought to be important in modulating the kinase activity[19–25]. In addition to ligand binding, the activity of chemoreceptors also depends on the level of receptor methylation that is controlled by the adaptation system. CheR preferentially methylates receptors that have inactive (kinase-inhibitory) conformation and CheB demethylates active receptors, respectively, increasing or decreasing their activity. These negative feedbacks enable the chemotaxis pathway to tune receptor activity dependent on the level of the background stimulation. Importantly, E. coli receptors are genetically encoded in a half-modified state, with two glutamines mimicking methylated glutamates. In the cell, receptors are organized into sensory arrays (clusters), where cooperative interactions between receptors result in amplification and integration of chemotactic signals[1,2].

Despite all this knowledge about the chemoreceptor structure, there is only limited mechanistic understanding of how receptors sense different environmental factors and transduce this information between individual domains. E. coli chemoreceptor Tar is one of the most studied bacterial sensors, which is known to mediate tactic responses to amino acids[26,27] but also to a number of other environmental stimuli, including sugars[28], metal ions[29], aromatic compounds[29,30], pH[31–33], temperature[34,35], and osmotic pressure[36,37]. Whereas amino acid attractants such as L-aspartate are known to bind directly to the ligand-binding site in the sensory domain[5], the detection mechanisms for most of the other stimuli remain unclear and at least some of these stimuli are likely to be sensed by the cytoplasmic part of the receptor[30,32,38]. Even for the conventional response to L-aspartate, pinning down the detailed mechanisms of signal transduction between individual receptor domains has proved to be difficult[2].

Another open question is how rapidly can the receptor response evolve under selection in novel environments. Although the aforementioned diversity of sensory domains illustrates receptor plasticity, the overall mechanism of chemoreceptor signaling is highly conserved, as evidenced by the functionality of hybrids that combine receptor domains from evolutionary distant species[4]. Nevertheless, the details of signal transmission can differ[39]. For example, in the case of Bacillus subtilis, the second-best studied model of bacterial chemotaxis, the sign of kinase activity regulation by chemoattractants is opposite. Whereas the binding of attractant inhibits CheA in E. coli, in B. subtilis attractant binding stimulates CheA. But how easily could such reversal of signaling happen through the course of evolution has not been established.

Mutations in different regions of receptors have been widely used to analyze individual aspects of receptor signaling, such as the regulation of the HAMP domain[13–15,23,40–42] and of the cytoplasmic signaling tip[43,44], as well as the control of signaling by the TM2 helix[16,45–48]. These mutations were observed to shift the equilibrium between active and inactive states of the receptor, thereby affecting its average basal activity and sensitivity to ligand stimulation, but not the sign of the response. Moreover, although several mutations have been described to invert responses to unconventional stimuli[49], including phenol[30], temperature[50,51],

oxygen[52], or cytoplasmic pH[32,53], none of these mutations inverted the canonical ligand response, and the interpretation of the observed response inversion remained unclear.

In contrast to these previous studies, here we demonstrate that sequence changes in the TM2 helix and in the region after the HAMP domain of Tar can easily invert its response to canonical amino acid ligands. Based on molecular dynamics (MD) simulations, we propose mechanisms behind this striking plasticity of bacterial receptors, thereby providing further insights into signal transduction between receptor domains. Finally, we demonstrate how Tar mutants with inverted signaling can be used as tools to identify receptor determinants that are responsible for detecting various types of environmental stimuli.

## Results

**Extension of TM2 inverts the response of Tar.** To better understand the mechanism of transmembrane signaling in E. coli chemoreceptors, we constructed a series of Tar mutants where the TM2 helix was shortened by one or two amino acids or extended by up to three amino acids (Fig. 1a, b). Signaling properties of these receptors were characterized using a reporter based on Förster (fluorescence) resonance energy transfer (FRET)[54,55]. This assay monitors the interaction between CheY fused to yellow fluorescent protein (CheY–YFP) and its phosphatase CheZ fused to cyan fluorescent protein (CheZ–CFP). Because this interaction requires CheY phosphorylation by CheA, increased CheA activity results in a stronger FRET signal and therefore a higher ratio of the YFP to CFP fluorescence.

These mutant receptors were first expressed from a plasmid in the strain VS181[54] that lacks all endogenous receptors but contains CheR and CheB adaptation enzymes. Under our growth and induction conditions (see Methods), the expression of Tar is several fold higher than its native expression level[55] and comparable to the total level of receptor expression in the wild-type E. coli. We observed that, except for Tar$^{TM2-2}$, all other constructs could still mediate responses to the amino acid ligand of Tar, α-methyl-D,L-aspartate (MeAsp) (Fig. 1b–d and Supplementary Fig. 1a–c). However, whereas cells expressing Tar$^{TM2+1I}$ or Tar$^{TM2-1}$ showed an attractant response that was similar to the wild-type Tar, constructs that carried two or three isoleucine insertions in TM2, Tar$^{TM2+2I}$ or Tar$^{TM2+3I}$, elicited an opposite, repellent response (Fig. 1c, d, Supplementary Fig. 1c, and Table 1). Interestingly, the apparent cooperativity of the observed response, characterized by the Hill coefficient of the fit to the data[54], was significantly lower for the inverted response (2.4 ± 0.3 for the wild-type Tar and 0.7 ± 0.1 for Tar$^{TM2+2I}$). This lower cooperativity might be related to the seemingly lower clustering efficiency of Tar$^{TM2+2I}$ (Supplementary Fig. 2).

This response inversion was further confirmed in a microfluidic assay[56] (see Methods and Supplementary Fig. 3). The receptorless strain UU1250[57] expressing the wild-type Tar

### Table 1 Responses of the wild-type and mutant Tar to MeAsp

| Tar protein | CheR⁺CheB⁺ strain | | cheRcheB strain | |
| --- | --- | --- | --- | --- |
| | **Response** | **EC$_{50}$$^a$ (μM)** | **Response** | **EC$_{50}$$^a$ (μM)** |
| Wild type | Attractant | 0.42 ± 0.03 | Attractant | 8.7 ± 0.2 |
| Tar$^{TM2-2}$ | No response | – | No response | – |
| Tar$^{TM2-1}$ | Attractant | 1.7 ± 0.5 | No response | – |
| Tar$^{TM2+1I}$ | Attractant | 3.8 ± 0.7 | No response | – |
| Tar$^{TM2+2I}$ | Repellent | 100 ± 14 | Repellent | 49 ± 4 |
| Tar$^{TM2+3I}$ | Repellent | 19 ± 2 | No response | – |

$^a$MeAsp concentration that inhibits or activates the kinase activity by 50%

showed the expected attractant response to MeAsp, with fluorescently labeled cells drifting up the MeAsp gradient in the observation channel of the device and also moving from the sink pore into the observation channel (Fig. 1e). In contrast, Tar[TM2+2I] mediated a repellent response, with cells drifting down the gradient of MeAsp and thus out of the observation channel into the sink pore (Fig. 1f).

In the adaptation-deficient *cheRcheB* background strain VH1[34], only the wild-type Tar and Tar[TM2+2I] exhibited a response to MeAsp (Table 1 and Supplementary Fig. 4), while other receptor mutants did not respond to MeAsp or to the established repellent NiCl$_2$[29]. The methylation system is thus apparently able to tune the modification state and thus activity of these receptors to an intermediate level, restoring their sensitivity to chemoeffectors. Importantly, however, changes in receptor methylation did not affect the sign of the response, since Tar[TM2+2I] exhibited a similar repellent response in both backgrounds.

To verify that the observed sign inversion is not a consequence of incorporation of the bulk side chains of two isoleucines in TM2, we constructed another mutant with two alanines inserted at the same position (Tar[TM2+2A]). Similar to Tar[TM2+2I], this mutant produced a repellent response to MeAsp (Supplementary Fig. 1d). Furthermore, Tar[V200IV201I] where two TM2 residues were mutated to isoleucines still retained the attractant response to MeAsp (Supplementary Fig. 1e), confirming that it is the length of TM2 and not specific sequence that is important to reverse the signal.

**Simulations suggest mechanism of the response inversion in the TM2 mutant**. To investigate how chemotactic responses can be inverted by the extension of TM2, we carried out MD simulations of the TM part of the wild-type and mutant Tar in lipid bilayers. Since the Tar dimer binds ligand with negative and half-of-the-sites cooperativity[58], here we compared the conformations of the TM helices in the ligand-free state in which both periplasmic ligand binding sites are free (apo-apo or AA) with the ligand-bound state in which a ligand is bound to one of the two binding sites in a dimer (apo-holo or AH). Our simulations used the previously established structural model of the receptor fragment[12], which combines the structure of a part of the periplasmic domain and the transmembrane helices of *E. coli* Tar with the structure of the HAMP domain of *Archaeoglobus fulgidus* receptor Af1503[9] (see Methods). Previous MD simulations of this model were consistent with experimental observations[12], and Tar with the native HAMP being replaced with Af1503 HAMP retained signaling capability[9], altogether suggesting that this truncated hybrid model can reproduce conformational changes within the full-length native Tar[22]. This structure was embedded into a lipid bilayer consisting of dipalmitoylphosphatidylcholine (DPPC) and simulated at 323 K, as widely applied for membrane proteins[7,12,59]. Notably, simulation time (800 ns) was apparently sufficient for the model structures of the wild-type and mutant Tar to equilibrate in either AA or AH states, as assessed by examining by the Cα root mean-square deviations (RMSD) from the initial structures as a function of time (Supplementary Fig. 5).

We observed that both the TM helices and the HAMP domain of the wild-type Tar adapted distinctly different conformations in the AA and AH states (Fig. 2a and Supplementary Fig. 6a). Conformations assumed by Tar[TM2+2I] were also different between the AA and AH states, but distinct from the corresponding wild-type structure (Fig. 2b and Supplementary Fig. 6a). Because the rigidity change of the junction between TM2 helix and the HAMP domain was suggested to be crucial for signal transmission[12–16], we calculated the average helicity of residues [211]GIRRMLLT[218] (all the residue numbers here and below correspond to the wild-type Tar) connecting TM2 and AS1

of the HAMP domain (Fig. 2c, Supplementary Fig. 7). These residues include the control cable[16] (GIRRM) and the first three residues (LLT) of AS1. We observed that for the wild-type Tar the helicity increased from 55% for the AA state to 74% for the ligand-occupied monomer of AH state (Fig. 2c). This agrees with the previous simulation-based model suggesting that the piston motion from α4 helix of periplasmic domain upon ligand binding results in slight bending of TM2 and enhances the rigidity and helicity of the control cable[12]. We further observed that for Tar[TM2-1] and Tar[TM2+1I] the simulated helicity of the junction region was only somewhat higher for the AA state compared to the wild-type Tar, and the helicity further increased upon transition to the AH state, apparently consistent with their wild-type like responses. In contrast, for Tar[TM2+2I] the simulated helicity of these junction residues showed a marked decrease in the ligand-occupied monomer, suggesting that this junction becomes less α-helical and rigid upon ligand binding. Interestingly, the average helicity of the junction residues in the ligand-free monomer of the AH state also slightly increased in wild-type Tar while decreasing in Tar[TM2+2I] as compared to the corresponding AA states (Supplementary Fig. 7), indicating that the ligand-free monomer may also be involved in signal transduction. We propose that the opposite change in helicity of the TM2–AS1 junction upon ligand binding might explain the response inversion in case of Tar[TM2+2I].

To ensure that these results are qualitatively independent the type of lipid, temperature, or on the length of the simulated receptor fragment, we repeated MD simulations for the wild-type Tar and Tar[TM2+2I] fragments that contain additional MH bundle connecting the HAMP domain (see below), with another widely used lipid, phosphatidylcholine (POPC)[25], and at 300 K (Supplementary Fig. 8). These simulations yielded qualitatively similar changes in helicity of the junction residues (Supplementary Fig. 9), confirming that our conclusions are indeed robust.

We additionally analyzed the time-evolution of secondary structure for the junction residues [211]GIRRMLLT[218] in detail using the DSSP program[60] (Supplementary Fig. 10). For wild-type Tar (Supplementary Fig. 10a), residues G211, I212, and R213 of the control cable adopt more stable α-helix structures upon ligand binding. L216 of AS1 also shows an increased helicity in the ligand-occupied monomer, consistent with a higher helicity of its junction region (Fig. 2c). For Tar[TM2+2I] (Supplementary Fig. 10b), ligand binding on the contrary largely decreases the helicity of these residues in the ligand-occupied monomer. Our results indicates that R213, R214, and M215 of the control cable generally favor helical structure, while stimulus input modulates helicity of most other junction residues, including G211 and I212 that were previously shown to be important for signaling in the control cable of chemoreceptor Tsr[15]. Interestingly, T218 adapted coil structure in all simulations, due to the kink on AS1 helix generated by P219. This might generally decrease the helicity of the residues L216 and L217, thus facilitating its modulation by stimulation both for the wild-type Tar and Tar[TM2+2I].

To further elucidate the details of the response inversion in Tar[TM2+2I], we analyzed the conformational changes of TM2 in the AA and AH states by calculating its bending angle[12] (Supplementary Fig. 11a, Supplementary Table 1) and the time evolution of helix curvature profile using HELANAL module[61] of MDAnalysis[62] (Supplementary Fig. 11b). We found that the bending angle of TM2 in the AA state of Tar[TM2+2I] is similar to the TM2 of the ligand-occupied monomer in the AH state of wild-type Tar. However, in the AH state of Tar[TM2+2I], the TM2 of the occupied monomer bends even further around the residue Leu205 near the end of TM2 (Supplementary Table 1, Supplementary Fig. 11b). The bending of TM2 breaks the helix structure around Leu205, which can be observed from the beginning of

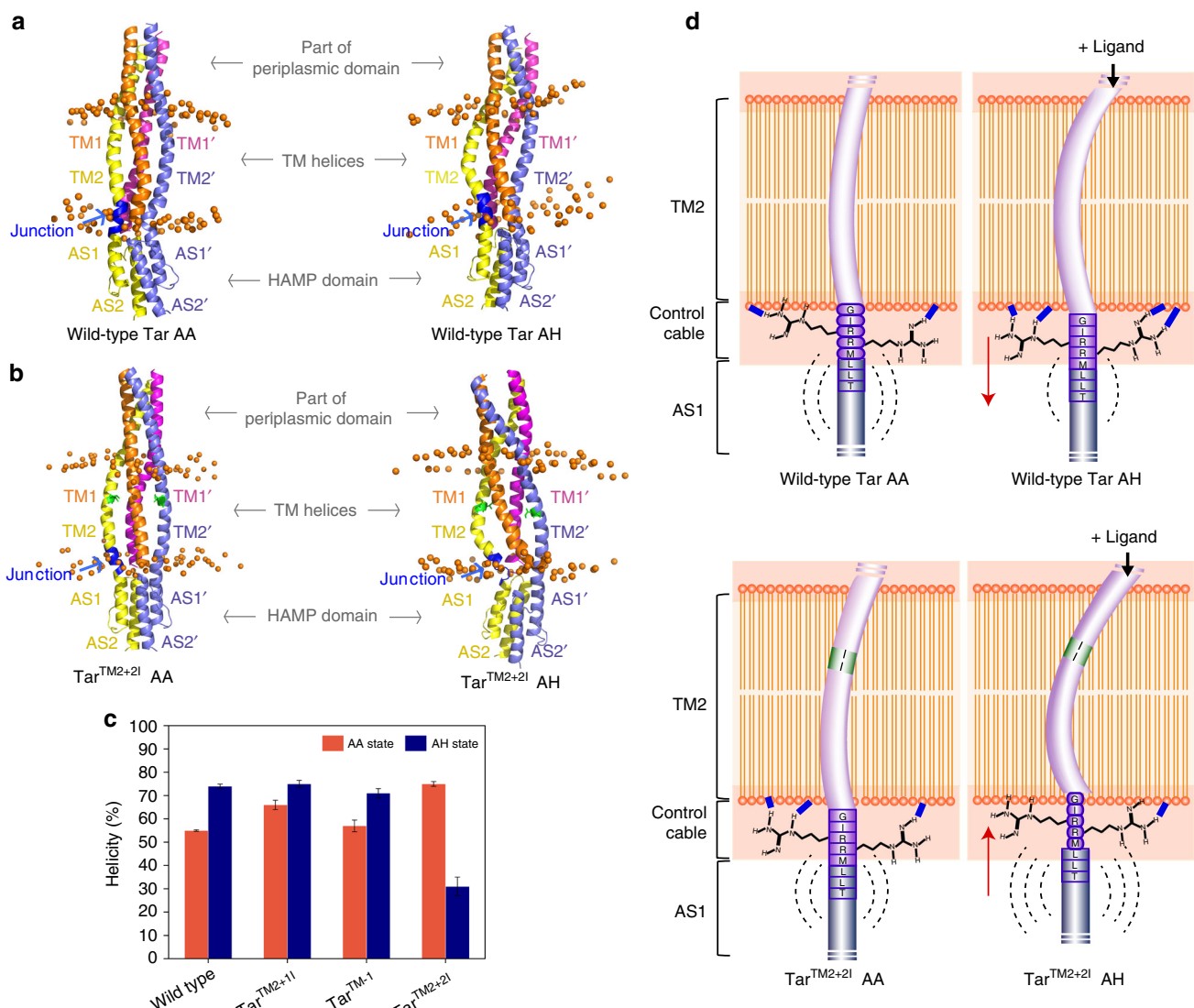

**Fig. 2** Mechanisms of signal inversion predicted by MD simulations. **a**, **b** Representative structures of AA (ligand-free) and AH (ligand-bound) models for wild-type Tar (**a**) and Tar$^{TM2+2I}$ (**b**) averaged from MD simulations. Notations for the helices in the TM and HAMP domains are shown along the structures. In AH states, TM1, TM2, AS1, and AS2 are from the ligand-occupied monomer, while TM1′, TM2′, AS1′, and AS2′ belong to the ligand-free monomer. Two isoleucines added in TM2 are shown in green in **b**. The TM2–AS1 junction residues $^{211}$GIRRMLLT$^{218}$ are shown in blue. DPPC phosphorus atoms are represented as orange spheres to show the solvent–membrane interfaces. The structures were prepared using PyMol. **c** The helicity of TM2–AS1 junction residues $^{211}$GIRRMLLT$^{218}$ was averaged from three independent MD simulations and compared between wild-type Tar, Tar$^{TM2-1}$, Tar$^{TM2+1I}$, and Tar$^{TM2+2I}$, shown as mean ± standard deviation. In the AH state, the helicity of the junction region from the ligand-occupied monomer was analyzed. **d** Schematic model of signal transduction to the HAMP domain and its inversion in Tar$^{TM2+2I}$. The ligand-promoted bending of TM2 triggers the inward shift of the control cable and AS1 in Tar and strengthens protein–lipid electrostatic interactions, which increase the helicity of the TM2–AS1 junction region and decrease the conformational dynamics of the control cable and the AS1 of HAMP domain. The outward shift of the control cable and AS1 in Tar$^{TM2+2I}$ induces opposite changes. The longitudinal displacement of the control cable and AS1 is highlighted using red arrow. The interactions of R213 and R214 are shown in blue lines. The conformational dynamics of the control cable and AS1 are represented in dash arcs

production simulations for Tar$^{TM2+2I}$. The respective bending angles of TM2 in the AA and AH states of wild-type Tar and Tar$^{TM2+2I}$ (Supplementary Table 1) are consistent with the helix curvature profiles (Supplementary Fig. 11b).

We also calculated the average longitudinal distance of the residues in the control cable and AS1 to the mass center of all phosphorus atoms at the lipid–cytoplasm interface (interface C, Supplementary Fig. 12a) in the AA and AH states of wild-type Tar and Tar$^{TM2+2I}$. When compared with the AA state, the ligand-promoted asymmetric bending of TM2 of the occupied monomer in the corresponding AH state results in a inward displacement of the control cable and AS1 in wild-type Tar

(Supplementary Fig. 12b, d). In contrast, the control cable and AS1 displacement in Tar$^{TM2+2I}$ is outward (Supplementary Fig. 12c, d). The movement of control cable is smaller than that of AS1 because the aromatic anchor residues in TM2 (Fig. 1a) and the protein–lipid interactions (Supplementary Fig. 13) fix the position of TM2 and hinder the further displacement of the control cable. The sliding of these residues is consistent with the longitudinal position of the residue Ile220 on AS1 in the ligand-occupied monomer relative to Ile220′ on AS1′ in the ligand-free monomer (Supplementary Fig. 12a, Supplementary Table 2), where Ile220 shows inward movement relative to Ile220′ in the AH state of wild-type Tar while moving outward in Tar$^{TM2+2I}$.

Consequently, whereas the movement of the control cable produces an inward sliding of AS1 relative to AS1′ in the wild-type Tar, it results in an outward relative sliding of AS1 in Tar$^{TM2+2I}$ (Supplementary Fig. 6b, c). Similar results were obtained by simulating longer structure in POPC at 300 K (Supplementary Table 2). The sliding of AS1 from the occupied monomer might change the side-chain interactions at the AS1–AS1′ interface in the HAMP dimer. No obvious helix rotation was observed in the HAMP domain along with the sliding of AS1, which is different from several previous reports[9,17] but consistent with other studies[10,23].

We next specifically explored the role of protein–lipid interactions in signaling to the HAMP domain. Because electrostatic interactions make major contribution to the protein–lipid interactions, we calculated the average electrostatic interactions between the TM2–AS1 junction residues $^{210}$YGIRRMLLTP$^{219}$ and the lipid bilayer in the AA and AH states of wild-type Tar and Tar$^{TM2+2I}$ (Supplementary Fig. 13a). In the wild-type Tar, the inward displacement of the control cable and AS1 in the ligand-occupied monomer stabilizes the protein–lipid interactions and consequently the helical structure of these junction residues (Fig. 2d). In contrast, the outward displacement of the control cable and AS1 in Tar$^{TM2+2I}$ changes the environment of the junction residues and destabilizes the protein–lipid interactions, thus decreasing the helicity of the junction region. Our simulations further suggest that changes of electrostatic interactions are mainly triggered by the interactions of residues R213 and R214 with the lipids. Moreover, the calculated electrostatic potential is consistent with the time average number of hydrogen bonds formed between these junction residues and the lipids, which increases in the AH state of wild-type Tar but decreases in the AH state of Tar$^{TM2+2I}$ (Supplementary Fig. 13b).

These changes further influence the conformational dynamics of the control cable and HAMP domain, as indicated by the analyzed root-mean-square fluctuations (RMSF, Supplementary Fig. 14, Supplementary Table 3). The control cable and AS1 in the AH state of wild-type Tar has lower RMSF than the AA state, and inverse is true for Tar$^{TM2+2I}$, indicating that ligand binding decreases the dynamics of the control cable and AS1 in wild-type Tar while increases their dynamics in Tar$^{TM2+2I}$. It is interesting that for the wild-type Tar simulations indicate an increased dynamics of AS2 in the AH state of wild-type Tar (Supplementary Fig. 14, Supplementary Table 3), which is opposite to the dynamic changes of AS1. Previous study proposed that the HAMP domain is stabilized in the kinase-off (ligand-bound) state, while it is destabilized in the kinase-on state[20,23,24]. Our study indicates that this stablization is limited to AS1. Finally, although chemotactic signals are initially triggered by the ligand-occupied monomer of Tar, the conformational dynamics of the control cable and AS1 in the ligand-free monomer change similarly to the ligand-occupied monomer (Supplementary Fig. 14), consistent with involvement of the ligand-free monomer in signaling.

To summarize, our results strongly suggest that the ligand-promoted bending of TM2 in wild-type Tar triggers an inward shift of the control cable and AS1 and strengthens the protein–lipid interactions, thereby stabilizing the helicity of junction region, decreasing the conformational dynamics of the control cable and AS1, and increasing the dynamics of AS2 (Fig. 2d). In Tar$^{TM2+2I}$, the further bending of TM2 instead triggers an outward shift of the control cable and AS1 and weakens the protein–lipid interactions, which decreases the helicity of junction region and increases the dynamics of the control cable and HAMP domain, thus eliciting an inverted response.

**Response inversion by mutations after the HAMP domain**. We next explored whether signaling could also be inverted by mutations in other regions of Tar, using the junction between the HAMP domain and the MH bundle as an example. To screen for such potential inversion mutants, we randomized five residues $^{267}$HVREG$^{271}$ connecting AS2 and the first methylation helix (MH1) (Fig. 3a). This receptor library was subsequently used for the selection of constructs that mediated spreading down the gradient of MeAsp established on a soft agar plate (Supplementary Fig. 15). The selection yielded constructs with the linker sequences $^{267}$GVPQM$^{271}$ (Tar$^{HAMP\_GVPQM}$), $^{267}$TLPRY$^{271}$ (Tar$^{HAMP\_TLPRY}$), and $^{267}$VVPAY$^{271}$ (Tar$^{HAMP\_VVPAY}$), all of which were confirmed to respond to MeAsp as a repellent in both of the FRET (Fig. 3b, c, Supplementary Fig. 16a, b) and microfluidic assays (Fig. 3d). The apparent cooperativity of inverted response mediated by Tar$^{HAMP\_GVPQM}$ was again lower that of the wild-type Tar (Fig. 3c), although these receptors seemed to show normal clustering (Supplementary Fig. 16c).

To elucidate the mechanism of inversion, we carried out MD simulations to compare the conformations of the linker region and the MH bundle for the AA (ligand-free) and AH (ligand-bound) states of the wild-type and mutant receptors. Simulations were performed for the receptor fragment containing a part of the periplasmic domain, the transmembrane helices, the HAMP domain, and the MH bundle. The models were embedded in DPPC or POPC bilayer and simulated at 323 or 300 K, respectively. The results showed that the conformations assumed by the wild-type and mutant Tar were different between the AA and AH states (Fig. 3e, Supplementary Figs. 17, 18). Notably, in all of these mutants the third position of the linker (position *e* of the first *a–g* heptad of MH1) was changed to proline, which generates a kink in the α-helix. Ligand binding resulted in helix bending around the linker region (Fig. 3e). As previous studies indicated that structural dynamics of the cytoplasmic domains might play an important role in modulating kinase activity[19–24], we analyzed the RMSF of the HAMP domain and MH bundle in both AA and AH states of the wild-type and mutant Tar receptors. The simulations using DPPC (Table 2, Supplementary Fig. 17, Supplementary Table 4) or POPC lipids (Supplementary Fig. 18, Supplementary Tables 4, 5) gave similar results. In the AH state of the wild-type Tar, the RMSF of AS1 in the HAMP dimer decreases, whereas that of AS2 and the four-helix MH bundle increases compared with the AA state. This is consistent with the previously proposed destabilization of the modification region in the kinase-off state[20,23,24]. In contrast, although the AH states of Tar$^{HAMP\_GVPQM}$, Tar$^{HAMP\_TLPRY}$, and Tar$^{HAMP\_VVPAY}$ showed a decreased RMSF of AS1 and AS2 of the HAMP domain, the dynamics of the MH bundle decreased in these cases.

We further analyzed the distance distributions of the residues on the HAMP domain and the MH bundle in the AA and AH states of wild-type Tar and Tar$^{HAMP\_GVPQM}$ (Supplementary Fig. 19a). For the HAMP domain of Tar$^{HAMP\_GVPQM}$ and AS1 of wild-type Tar, the distance distributions of the residue pairs become slightly narrower in the AH state than the AA state (Supplementary Fig. 19b, d), consistent with its lower RMSF upon ligand binding. In contrast, the distance distributions of the residue pairs on the MH1 and MH2 change oppositely in wild-type Tar and Tar$^{HAMP\_GVPQM}$, with wild-type Tar having wider distributions in the AH state while Tar$^{HAMP\_GVPQM}$ having narrower distributions (Supplementary Fig. 19c, e), in line with the higher RMSF data.

We propose that the linker region in the wild type and mutants propagate opposite conformational effects on the dynamics of the MH bundle, which might explain the response inversion in the mutants. The piston force from the periplasmic domain and TM2 upon ligand binding changes the AS1–AS1′ interface the dynamics of the HAMP domain. In the wild-type Tar, there is

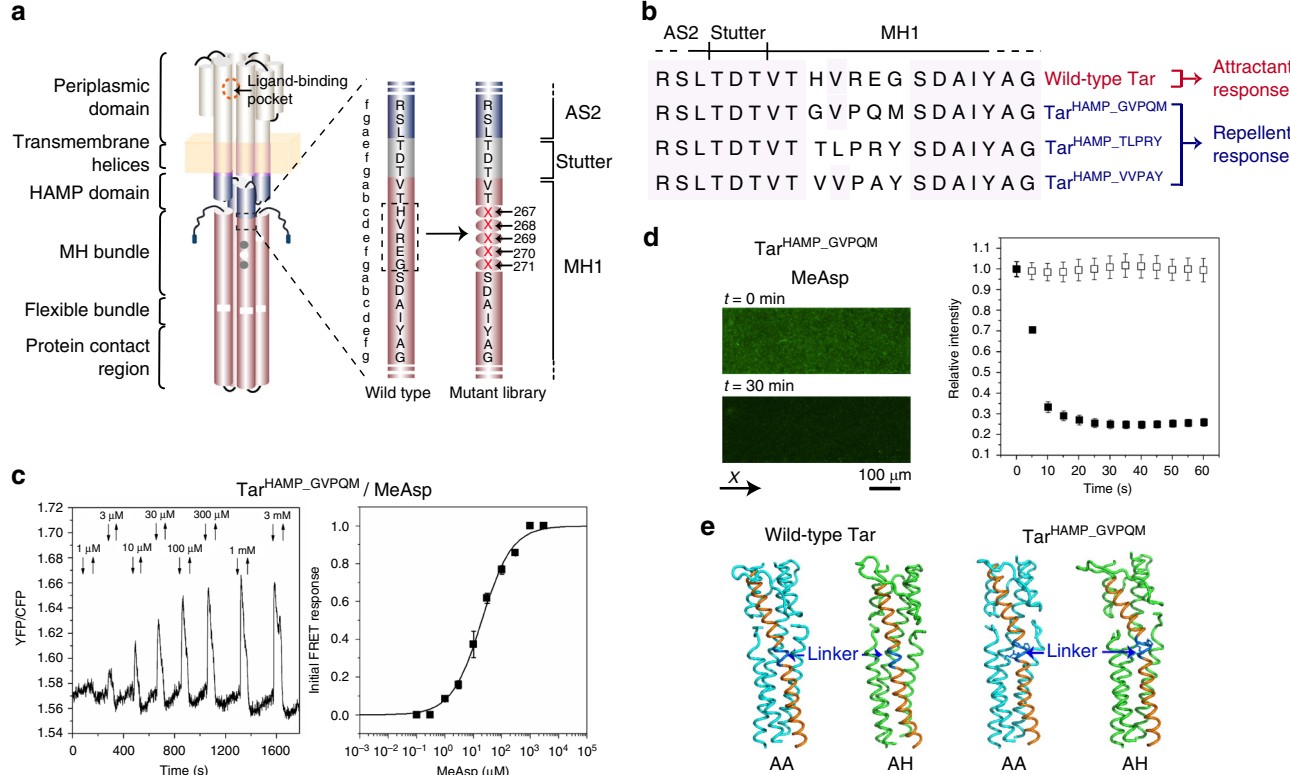

**Fig. 3** Repellent responses of Tar with mutations after the HAMP domain. **a** Schematic representation of the Tar[1–266]-XXXXX-[272–553] library with a random linker region made after the HAMP domain of Tar. **b** Sequence alignment for the wild-type and mutant Tar receptors. The receptors that induced attractant or repellent responses to MeAsp are indicated. **c** FRET measurements and dose response of *E. coli* CheR$^+$CheB$^+$ strain VS181, expressing Tar$^{HAMP\_GVPQM}$ as a sole receptor, as well as the FRET pair, to a stepwise addition and subsequent removal of the indicated concentrations of MeAsp. Error bars indicate standard deviation of three independent biological replicates; note that for most data points error bars are smaller than the symbol size. EC$_{50}$ of the dose response is 19.6 ± 1.7 μM. Hill coefficient is 0.8 ± 0.1. **d** Microfluidic assay of the chemotactic response of *E. coli* strain UU1250 expressing Tar$^{HAMP\_GVPQM}$ to MeAsp. Cell distribution in the observation channel was recorded when the source was 10 mM MeAsp for Tar$^{HAMP\_GVPQM}$ (scale bar, 100 μm). The x-component (black arrow) indicates the direction up the concentration gradient. Relative cell density (fluorescence intensity) in the observation channel for Tar$^{HAMP\_GVPQM}$ responding to MeAsp (closed squares) or without MeAsp (open squares) were plotted over time. The cell density in the observation channel before MeAsp stimulating (t = 0) was normalized to be one. Error bars indicate standard deviation of three independent biological replicates. **e** Representative structure of AA (ligand-free; cyan) and AH (ligand-bound; green) models of the wild-type Tar and Tar$^{HAMP\_GVPQM}$ simulated in the DPPC lipids. Only part of the HAMP domain and the MH bundle are shown. The linker region is shown in blue and the helices of AS2 and MH1 from the occupied monomer in the AH state and in the ligand-free AA state are shown in orange

**Table 2 Structural dynamics of the MH bundle in the AA and AH states of the wild-type and mutant Tar dimer**

| Receptor | RMSF of MH bundle[a] | |
|---|---|---|
| | AA (Å) | AH (Å) |
| Wild-type Tar | 1.24 ± 0.22 | 1.35 ± 0.17 |
| Tar$^{HAMP\_GVPQM}$ | 1.41 ± 0.20 | 1.28 ± 0.19 |
| Tar$^{HAMP\_TLPRY}$ | 1.35 ± 0.22 | 1.25 ± 0.18 |
| Tar$^{HAMP\_VVPAY}$ | 1.38 ± 0.16 | 1.21 ± 0.14 |

[a]The values were averaged from the RMSF plots of the residues 267–297 of MH1 and MH1′ and the residues 485–509 of MH2 and MH2′ for the four-helix MH bundle in three MD simulation trajectories, shown as mean ± standard deviation.

a support force from the linker region and MH1 to balance this piston force, which leads to the destabilization of MH1. In the mutant receptors, however, R269P substitution in the linker region generates a helix kink that breaks the balance between piston force and support force and results in stabilization of the MH bundle.

To test whether the R269P substitution alone is sufficient for signal inversion, we introduced this mutation into the wild-type Tar. However, Tar$^{R269P}$ showed no response to MeAsp, possibly

because P269 (position e) and G271 (position g) together broke the MH1 helix thus fully inhibiting receptor activity. We hypothesized that additional substitution of G271 at the helix–helix interface to hydrophobic residues Y or M, as observed in all three functional mutants ($^{267}$GVPQM$^{271}$, $^{267}$TLPRY$^{271}$, and $^{267}$VVPAY$^{271}$) might enhance helical packing and restore activity. Indeed, FRET measurements showed that Tar$^{R269PG271Y}$ and Tar$^{R269PG271M}$ recovered the ability to sense MeAsp as a repellent (Supplementary Fig. 20). This suggests that, besides the kink introduced by R269P, the inverted response also requires hydrophobic substitutions at G271 that stabilize helical packing.

**Mapping sensory determinants for environmental stimuli.** Many bacterial chemotaxis receptors and sensory histidine kinases are able to respond to a variety of very different environmental factors, which are unlikely to be sensed by the periplasmic sensory domains. In most cases the mechanisms and even the respective sensory regions of receptors remain unclear. We hypothesize that mutants with an inverted response can be used to map the sensory elements of receptors that are responsible for detection of these stimuli. According to positions of the inversion sequences, Tar can be subdivided into three parts (Fig. 1a): (I) TM1, the periplasmic domain and a part of TM2; (II)

the HAMP domain and its junction with TM2; and (III) the region below the HAMP domain. Assuming that conformational changes in the receptor that are elicited by all stimuli are similar, inverted response to a particular stimulus for both $Tar^{TM2+2I}$ and $Tar^{HAMP\_GVPQM}$ indicates that this stimulus is detected by the region I. If only $Tar^{HAMP\_GVPQM}$ but not $Tar^{TM2+2I}$ response is inverted, the stimulus is likely to be detected by the region II, and if no inversion is observed the response must be mediated by the region III.

As the control for this approach, we confirmed which sensory region is responsible for the detection of the external pH. Tar is known to mediate an attractant response to low pH and a repellent response to high pH, primarily through the periplasmic domain[31,38]. Consistent with this mode of sensing by region I, cells expressing $Tar^{TM2+2I}$ or $Tar^{HAMP\_GVPQM}$ as a sole receptor both showed an inverted response to pH (Fig. 4a).

We next mapped sensing regions for a number of other stimuli. Divalent cations $Ni^{2+}$ and $Co^{2+}$ are established repellents for *E. coli* Tar[29], but the mechanisms of their sensing remain elusive. While it was initially suggested that $Ni^{2+}$ is sensed by the periplasmic domain of Tar via the periplasmic nickel-binding protein[63], this mode of sensing was challenged by a subsequent study[64]. We observed that the response of $Tar^{TM2+2I}$ to $Ni^{2+}$ was similar to the wild-type Tar whereas the response of $Tar^{HAMP\_GVPQM}$ was inverted (Fig. 4b). This suggests that $Ni^{2+}$ is detected by the region II, most likely by the HAMP domain, and not by the periplasmic domain. This could be further supported by the analysis of the Tar mutant that completely lacks the periplasmic domain ($Tar^\circ$-T303I)[65], which still showed a repellent response to $Ni^{2+}$ (Supplementary Fig. 21).

We also investigated the sensing of aromatic compounds toluene and *o*-xylene, which are attractants for Tar (Fig. 4c, Supplementary Fig. 22a). Both $Tar^{TM2+2I}$ and $Tar^{HAMP\_GVPQM}$ produced attractant responses to toluene and *o*-xylene (Fig. 4c, Supplementary Fig. 22a), suggesting that the main sensing determinant for these compounds resides within the region below the HAMP domain. Nevertheless, the response of $Tar^{HAMP\_GVPQM}$ was weaker than the wild-type Tar, indicating that other receptor segment(s), such as TM helices[30] might also be involved in detecting these compounds.

*E. coli* cells further show repellent responses to osmotic stress, which apparently enables them to avoid regions of high osmolarity[36,37]. Because all of the wild-type and mutant receptors showed repellent responses to non-ionic osmolytes betaine, sucrose, and L-proline (Fig. 4d, Supplementary Fig. 22b, c), we concluded that osmotic stress is primarily sensed by the region below the HAMP domain. In contrast, for the ionic osmolyte NaCl the wild-type Tar showed an attractant response in the tested range of concentrations, whereas both of the mutants produced repellent responses (Fig. 5a–c). These results suggest that Tar-mediated sensing of NaCl primarily occurs via the periplasmic domain and is thus different from the general osmotic response. To elucidate the underlying mechanism, we performed MD simulations on the structure of the Tar periplasmic domain at 50 mM NaCl. The systems remain stable during simulations as indicated by low Cα RMSD values (Supplementary Fig. 23a). We subsequently performed the backbone-RMSD clustering for the conformations of final 800 ns with a cutoff of 2.0 Å and compared the structures of the two mostly populated clusters, 60.4% at 0 mM NaCl and 75.8% at 50 mM NaCl. This comparison shows that NaCl induces a distinct conformation of the periplasmic domain that includes a 3.8 Å inward sliding of the α4 helix relative to the α1 helix, as well as relative rotation of the two monomers for 25.5° (Fig. 5d). These two modes of the conformational change are in line with the rearrangement triggered by binding of aspartate[5,6]. We further

hypothesized that other salt ions might elicit similar effects on the periplasmic domain of Tar. Consistently, the wild-type Tar showed an attractant response to KCl, while both mutants mediated the repellent response (Supplementary Fig. 22d).

## Discussion

Despite bacterial chemotaxis pathway being one of the most conserved and best studied signaling systems in bacteria, many aspects of stimulus sensing and signal transduction through bacterial chemoreceptors remain enigmatic. In this study, we report that mutations in specific regions of *E. coli* chemoreceptor Tar can invert its response to ligands, illustrating striking plasticity of bacterial sensors. These results suggest that under selection bacterial sensors can easily change not only their ligand specificity but also the sign of their response. Furthermore, the obtained response-inverting mutations enabled us to better understand the mechanism of signal transmission through chemoreceptors and to map sensory determinants for unconventional chemotactic stimuli.

For conventional ligands that bind to the sensory domains of chemoreceptors, signal transmission is known to occur through conformational changes in individual receptor domains. Nevertheless, these conformational signals are very subtle and elucidating their transmission between domains remains a challenge. This is particularly true for the input and output signaling of the HAMP domain that has a central role in signal transmission in chemotaxis receptors, as well as in a wider class of bacterial sensory histidine kinases[9,10,42,66,67]. The mode of signaling by unconventional physico-chemical stimuli is even less clear. These stimuli are unlikely to all directly affect the sensory domain, and for most of them neither the sensory mode nor the mechanism of signaling have been established.

Mutagenesis has been previously extensively used to investigate the function of bacterial chemoreceptors, and multiple mutations in different receptor regions are known to have either activating or inhibitory effects on receptor activity[13–16,23,40–48]. For example, a previous study showed that insertion or deletion of one residue in TM2 or the control cable of Tsr resulted in kinase-off or kinase-on output shifts[14]. Similar shifts were observed upon repositioning the Trp209/Tyr210 pair in TM2 of Tar[47]. However, none of these previously described mutations inverted the response to canonical chemical ligands. And although several mutations that invert responses to unconventional stimuli[49], such as phenol[30], temperature[50,51], oxygen[52], or cytoplasmic pH[32,53], have been described, the molecular mechanism of signaling by these non-canonical stimuli has never been elucidated, and therefore the interpretation of the observed response inversion remained unclear. Moreover, these previously reported mutations did not invert the canonical ligand response, thus failing to provide mechanistic insights into receptor signaling.

Here, we show that the response of bacterial receptor Tar to the canonical amino-acid attractants can be inverted by perturbing the sequences above or below the HAMP domain. Because signaling by canonical stimuli is much better understood, we propose molecular mechanisms explaining these inversions and thereby provide deeper insights into signaling through the native receptors. The mechanisms of the observed inversions could be explained by MD simulations, providing insights into signal transmission through the HAMP domain. For the response inversion by mutations above HAMP, our analysis suggests that TM2 extension further increases the bending angle of the TM2 helix. This in turn decreases the helicity of the TM2–AS1 junction region $^{211}$GIRRMLLT$^{218}$ and elicits an outward shift of the control cable (GIRRM; part of this junction region) and AS1 helix of HAMP. Our explanation is consistent with the previous model

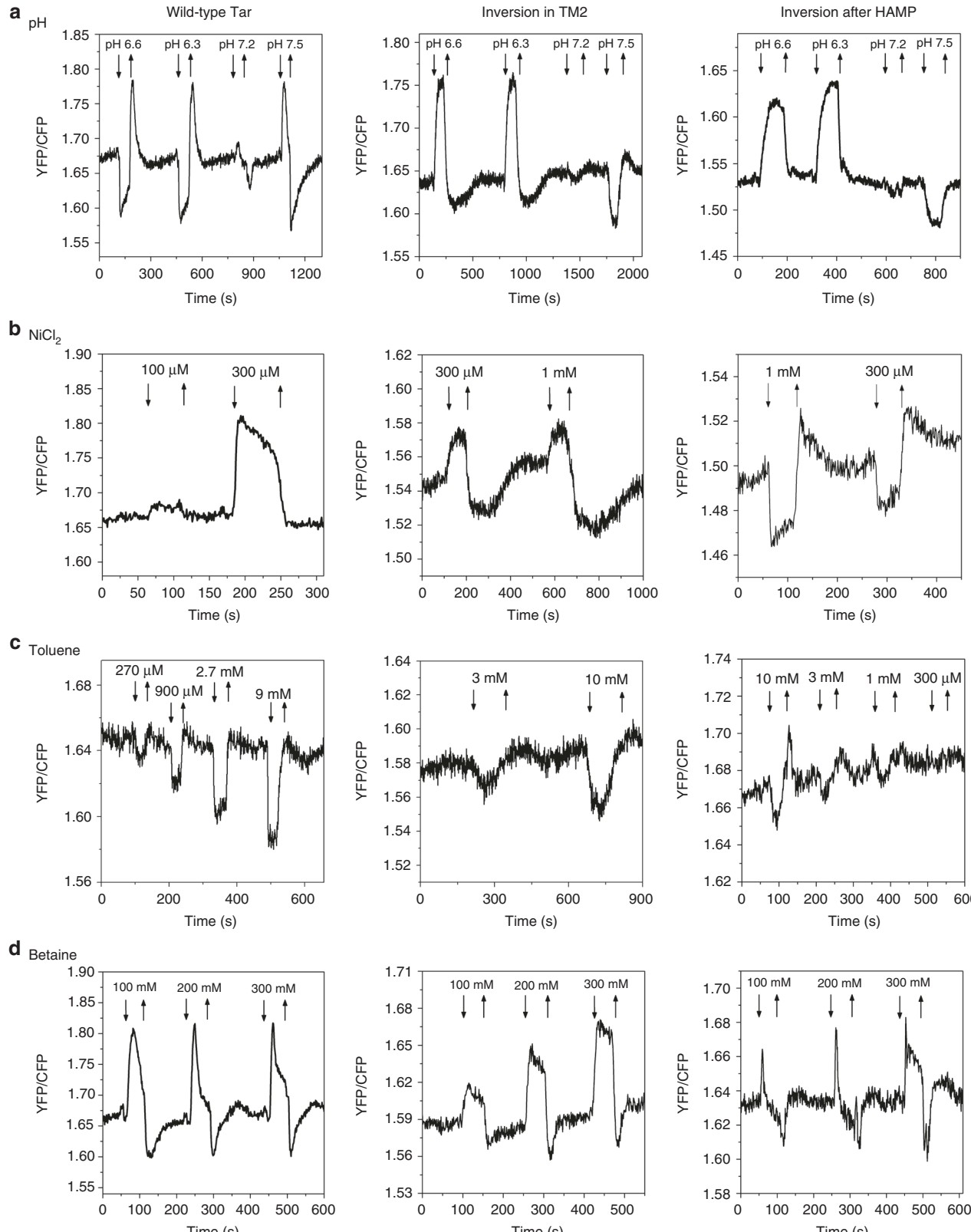

**Fig. 4** Responses of the wild-type and mutant Tar to external pH, NiCl$_2$, toluene, and betaine. **a** FRET measurements of external pH responses in the CheR$^+$CheB$^+$ *E. coli* strain VS181 that expresses only the wild-type Tar, Tar$^{TM2+2I}$ (inversion in TM2), or Tar$^{HAMP\_GVPQM}$ (inversion after HAMP). Cells were pre-adapted in the buffer at neutral pH (7.0) prior to stimulation with decrease or increase in pH. **b–d** FRET responses to a stepwise addition (down arrow) and subsequent removal (up arrow) of the indicated concentrations of NiCl$_2$ (**b**), toluene (**c**), or betaine (**d**)

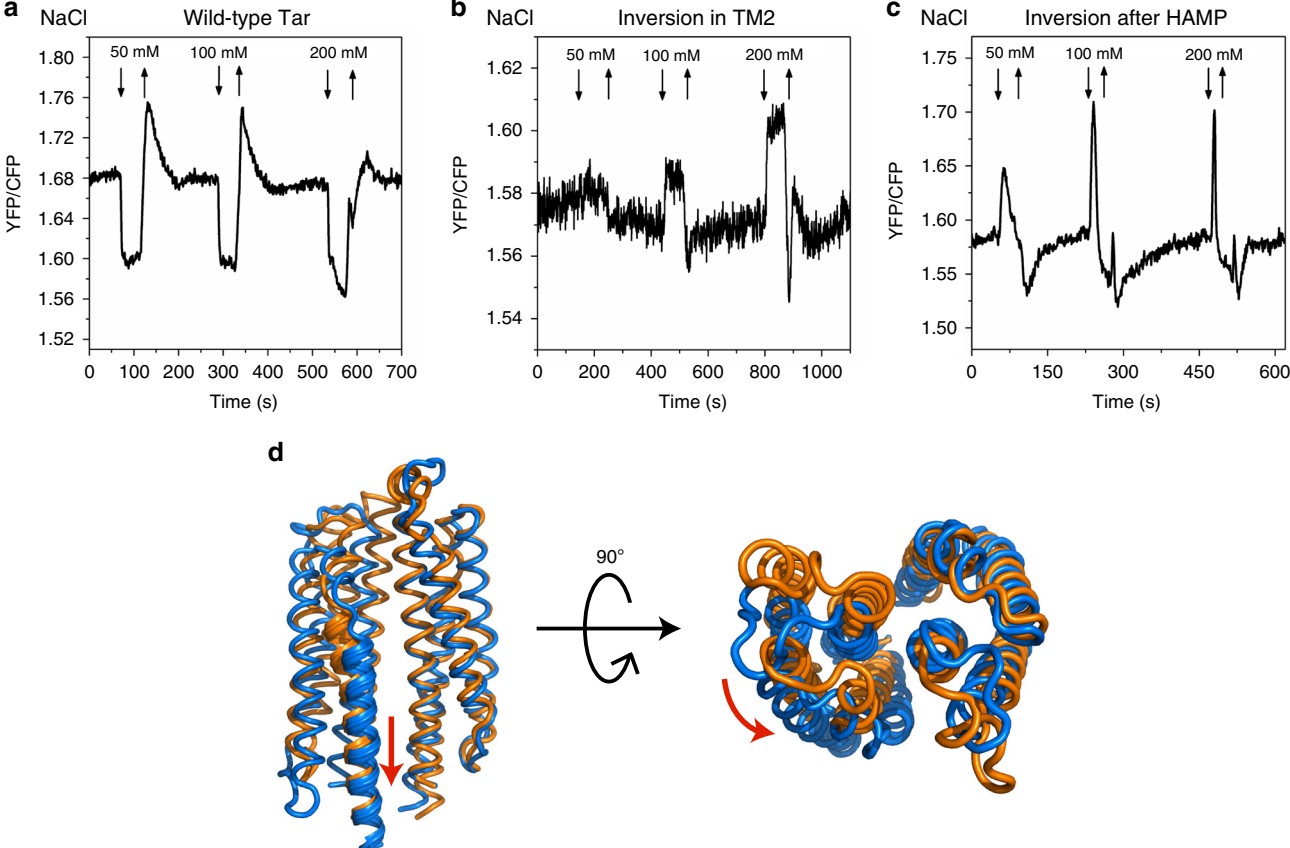

**Fig. 5** Responses of the wild-type and mutant Tar to NaCl. **a–c** FRET response of buffer-adapted *E. coli* CheR$^+$CheB$^+$ strain VS181 expressing the wild-type Tar, Tar$^{TM2+2I}$ (inversion in TM2), and Tar$^{HAMP\_GVPQM}$ (inversion after HAMP) as a sole receptor to a stepwise addition (down arrow) and subsequent removal (up arrow) of the indicated concentrations of NaCl. **d** Representative conformations of Tar periplasmic domain, simulated at 0 (orange) or 50 mM (blue) NaCl, with the side and top view. The structures were aligned with one monomer of the dimer. The red arrows describe the inward sliding of the α4 relative to α1 helix and rotation of one monomer relative to the other at 50 mM NaCl. Residues 155–181 of the periplasmic domain are highlighted to indicate the inward movement of α4 helix

suggesting that the rigidity change of the junction between the TM helices and the HAMP domain is crucial for the modulation of transmembrane signaling[12–16]. Our results also demonstrate that the region connecting AS2 and MH1 is crucial for the signal output of Tar HAMP domain. Mutations in this linker region apparently invert the dynamic changes in the MH bundle structure that are induced upon ligand binding, stabilizing the MH bundle rather than increasing its dynamics as for the wild-type receptor[20,22–24]. In these mutant receptors, the kink breaks the balance between the piston force and the support force from the linker and MH1, leading to decreased dynamics of the MH bundle. We show that substitutions R269P and G271Y or G271M in the mutated linker are sufficient for eliciting this opposite response, with R269P producing a kink in the helix and G271Y or G271M likely enhancing helical packing to stabilize receptor activity.

Our results are thus consistent with the view that receptor domain segments have different dynamics in the kinase-on and kinase-off signaling states[19–24] and provide experimental evidence to support these models[20,23,24]. Overall, ligand binding to the wild-type Tar generates the piston force from the periplasmic domain, which results in bending of TM2[12], reduces fluctuations of the HAMP domain and increases dynamics of the MH bundle[20,22–24] (Fig. 2d). Importantly, our analysis enables several further refinements of this model of signal transmission. It suggests that, for the wild-type Tar, TM2 sliding induced by ligand binding promotes interactions of the junction residues, particularly of R213 and R214,

with the membrane. Along with sliding of the control cable and AS1, these changes stabilize helical conformation of the junction residues and AS1 of the HAMP domain. In contrast, the dynamics of AS2 rather increases, possibly propagating this change to the MH bundle. Notably, similar conformation changes are observed for both the ligand-occupied and ligand-free monomers, suggesting the involvement of the latter in signaling. The proposed involvement of protein–lipid interactions in receptor signaling is consistent with signaling effects of mutations in I214 and K215 of Tsr, which are likely to affect electrostatic interactions of the control cable with lipids[15].

As a side observation, we found that in both cases the inverted response was apparently less cooperative compared with attractant response mediated by the wild-type Tar. Although this finding requires further investigation, at least in case of Tar$^{TM2+2I}$ lower cooperativity seems to be consistent with less efficient receptor clustering, which might weaken cooperative interactions between receptors. The effect of mutations in the linker region after HAMP on cooperativity of receptor interactions appears to be more subtle, since Tar$^{HAMP\_GVPQM}$ form clusters that similar in size to those of the wild-type Tar.

Besides insights into receptor signaling, we used inversion mutants to determine Tar regions responsible for detection of unconventional tactic stimuli (Supplementary Fig. 24). Detection of Ni$^{2+}$, and presumably of other divalent cations, is apparently mediated by the receptor region that includes HAMP domain and its junction with TM2. This contradicts to the earlier model of

$Ni^{2+}$ sensing through interactions between the periplasmic sensory domain of Tar and the nickel-binding protein NikA[63], but it agrees with the subsequent study[64] showing that the NikA, NikB, and NikC in the nickel uptake system are not necessary for $Ni^{2+}$ detection. Our results suggest that $Ni^{2+}$ acts intracellularly, being transported into the cell either by the NikABC or by other transport systems[68].

Sensing of aromatic compounds is apparently mediated by both TM helices and the region below the HAMP domain. Although the involvement of the TM helices in phenol sensing was shown before[30], our results suggest that the region below the Tar HAMP domain plays a dominant role in the detection of aromatic compounds.

Finally, the response to hyperosmotic shock elicited by non-ionic osmolytes is mediated by the region below the HAMP domain. This is apparently consistent with the previously reported osmolarity-induced relative movement of Tar dimers within the trimer structure, which occurs around the cytoplasmic tip of receptors[36]. Nevertheless, osmolarity might also induce conformational changes in the region below the HAMP as suggested for the osmosensor EnvZ[69]. In contrast to this general osmotic response, sensing mechanism for ionic osmolytes NaCl and KCl involved the periplasmic domain of Tar. We propose that monovalent salts screen the coulombic interactions at the surface of Tar periplasmic domain, thereby reshaping its conformation and dynamics (Supplementary Fig. 23b). Similar mechanism of salt ion detection might be used by the two-component histidine kinases[70].

## Methods

**Strains and plasmids**. *E. coli* strains and plasmids used in this work are listed in Supplementary Table 8. For the FRET measurements, receptorless *cheY cheZ* strain VS181 [Δ(*cheYcheZ*)Δ*aer*Δ*tsr*Δ(*tar-tap*)Δ*trg*][54] or receptorless *cheR cheB cheY cheZ* strain VH1 [Δ(*cheRcheB*)Δ(*cheYcheZ*)Δ*aer*Δ*tsr*Δ(*tar-tap*)Δ*trg*][34] were transformed with the plasmid pVS88 expressing the FRET pair CheY-YFP/CheZ-CFP, and a plasmid expressing the receptor of interest. The receptorless strain UU1250 [Δ*aer*Δ*tsr*Δ(*tar-tap*)Δ*trg*][57] was used for the soft agar plate assay, microfluidic assay, and fluorescence imaging.

**Molecular cloning and mutagenesis**. To construct the plasmid pSB13, the coding sequence of Tar was amplified by PCR, digested using *Nde*I and *Bam*HI, and ligated with the plasmid pKG116. The plasmids pSB14 to pSB20 were generated by Q5® Site-Directed Mutagenesis Kit (New England BioLabs) using the plasmid pSB13 as the template. For the library of Tar[1–266]-XXXXX-[272–553], the coding sequences of the receptor fragments with random linkers were amplified and inserted between the restriction sites *Nde*I and *Bam*HI of the plasmid pKG116 using circular polymerase extension cloning (CPEC)[71].

**FRET measurements**. Cells with the plasmids encoding the FRET pair CheY-YFP/CheZ-CFP and a receptor of interest were grown in tryptone broth (TB; 1% tryptone and 0.5% NaCl) supplemented with antibiotics (100 μg ml$^{-1}$ ampicillin; 17 μg ml$^{-1}$ chloramphenicol) overnight at 30 °C. The overnight culture was diluted 1:100 in 10 ml of fresh TB medium supplemented with 100 μg ml$^{-1}$ ampicillin; 17 μg ml$^{-1}$ chloramphenicol, 50 μM IPTG, and 2 μM salicylate. After growth to $OD_{600}$ of 0.6 at 34 °C and 275 r.p.m., cells were harvested and washed twice with the tethering buffer (10 mM $KH_2PO_4$/$K_2HPO_4$, 0.1 mM EDTA, 1 μM methionine, 10 mM sodium lactate, pH7.0). Under these conditions, the expression level of Tar in receptorless cells is ~3.6-fold higher than the native expression level of Tar[55] and thus comparable to the total level of receptor expression in wild-type cells. FRET measurements were performed as described before on an upright fluorescence microscope (Zeiss Axio Imager.Z1)[31]. Briefly, cells were attached to a polylysine-coated coverslip and placed into a flow chamber. Under a constant flow (0.5 ml min$^{-1}$), cells were adapted in the tethering buffer and subsequently stimulated with specified stimuli. Fluorescence signals were continuously recorded in the cyan and yellow channels using photon counters with a 1.0 s integration time. Data were plotted and analyzed as described before[31].

**Soft agar plate assay**. Minimal A agar[4] supplemented with antibiotics and inducers was used for selecting functional Tar mutant from the Tar[1–266]-XXXXX-[272–553] library. Chemical solution was applied as a line to the center of the plate and incubated at 4 °C for 16 h to generate a chemical gradient. Overnight culture expressing the library was applied to the plate at a defined distance from the center, and plates were incubated at 30 °C. Cells that migrated the farthest in the chemical gradient were re-inoculated on a new plate for the second round of selection. Receptor-expressing plasmids for the best-chemotactic cells were isolated and re-transformed into UU1250 to confirm the mutant phenotype. The sequence of the linker in the selected receptor mutants was identified by DNA sequencing.

**Microfluidic assay**. *E. coli* UU1250 cells expressing mutant receptors and GFP were grown at 34 °C in TB supplemented with 100 μg ml$^{-1}$ ampicillin; 17 μg ml$^{-1}$ chloramphenicol, 100 μM IPTG, and 2 μM salicylate until the $OD_{600}$ reached 0.6. Cells were harvested by centrifugation and washed twice with the tethering buffer. The responses of *E. coli* cells to concentration gradient of MeAsp were measured using a microfluidic device described previously[56] (Supplementary Fig. 3). For preparation of the device, 4% agarose was added at the source side pore to seal the interface with the observation channel. 10 μl tethering buffer each was then added into the source and sink side pores. Afterwards, *E. coli* cells were added into the sink pore to a final $OD_{600}$ of 2.0 and allowed to diffuse into the observation channel for 1 h. Compound solution was then added to the source pore and allowed to gradually diffuse through the agarose gel into the observation channel. Cell fluorescence in the observation channel was measured over time, starting immediately after compound addition, using Nikon Ti-E inverted fluorescence microscope with a ×20 objective lens and Lumencor SOLA-SEII equipped with Andor Zyla sCMOS camera. Data were analyzed using ImageJ (Wayne Rasband, National Institutes of Health, USA).

**Fluorescence imaging**. Receptorless *E. coli* strain UU1250 expressing receptors of interest together with the catalytically inactive YFP-CheR$^{D154A}$ as a marker were applied to a thin agarose pad (1% agarose in tethering buffer). Fluorescence images were recorded on a Nikon Ti-E inverted fluorescence microscope with Lumencor SOLA-SEII equipped with Andor Zyla sCMOS camera. Each imaging experiment was performed in duplicate on independent cultures.

**MD simulations**. Details on structure construction and MD simulations for the wild-type and mutant Tar receptors are described in Supplementary Methods and Supplementary Tables 7–10. The model used for simulations is based on previous work[12] and combines the structures of periplasmic domain, TM helices, and MH bundle from *E. coli* Tar with the structure of the HAMP domain from *A. fulgidus* receptor Af1503. The *A. fulgidus* structure[9] was used because the structure of the native HAMP domain of Tar is neither available nor can be modeled with quality sufficient for MD simulations. The models of the simulated systems were embedded in a lipid bilayer with DPPC simulated at 323 K[7,12,59] or with POPC[25] simulated at 300 K. The used structural restraints are summarized in Supplementary Tables 8–10. The first 200 ns for all trajectories were regarded as equilibration and not used for analysis. All simulations were performed at supercomputer DRACO of Max Planck Computing and Data Facility (MPCDF).

## Data availability

The data supporting the findings of the study are available in this article and its Supplementary Information files, or from the corresponding author upon request.

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

## Acknowledgements

This work was partly supported by grant 294761-MicRobE from the European Research Council.

## Author contributions

S.B., F.J., and V.S. designed research; S.B. and F.J. performed research and analyzed data; and S.B., F.J., and V.S. wrote the paper.

## Additional information

**Competing interests:** The authors declare no competing interests.

