## [Peer Review File · Nature Communications]

Reviewers' comments:

Reviewer #1 (Remarks to the Author):

In this manuscript, Bi et al., report characterization of mutant aspartate chemoreceptors that show inverted kinase control responses to various stimuli. Two types of inverted mutants were studied: (1) Receptors with 2-3 additional residues in the membrane-spanning TM2 helices that couple conformational changes in the ligand-binding domain to those in the cytoplasmic HAMP domain. (2) Receptors with randomized five-residue sequences near the junction of the HAMP domain helices and the methylation helices that control kinase output and sensory adaptation. The mutant receptors were cleverly used to determine what part of the molecule was likely responsible for sensing various less well understood stimulus inputs. The mutant receptor behaviors are quite interesting and shed new light on transmembrane and intracellular signaling by this well-studied family of bacterial chemoreceptors.

Molecular dynamics simulations of portions of the mutant receptors provided mechanistic explanations for the signal inversion. However, I have concerns about the validity of the structural models used for most of the MD simulations. I do not feel that those simulations make a convincing case for the mechanisms proposed. I offer a number of suggestions for improving the solidity of the scientific case and the clarity of the presentation.

major issues:

(1) The model for the MD simulations of the periplasmic domain-TM bundle-HAMP bundle was essentially the one used by Park et al (2011) in an earlier MD study. It employs a chimeric protein in which part of the periplasmic domain and the TM bundle of Tar are joined to a HAMP domain (Af1503) from a hyperthermophile. From the supplementary methods, it appears that the simulations were then carried out at 50°C. Thus, part of the model protein evolved to operate 15-20 degrees below the simulation temperature and part evolved to operate at 20-30 degrees above the simulation temperature. Given the conflicting structural and dynamic interactions between those two disparate parts, it's difficult to see how that model could reliably reproduce what happens in the native receptor at its normal operating temperature. Some of the constraints imposed on the models of the mutant proteins (e.g., wild-type piston displacements and maintenance of TM2 helicity) might also have influenced the simulation outcomes. (It was unclear exactly what structural constraints, if any, were placed at the C-termini of the HAMP helices.)

(2) The model for MD simulations of the HAMP-methylation helix interactions was based on the crystal structure of a chimeric protein containing the Af1503 HAMP domain joined to the signaling domain of Tsr, an E. coli receptor closely related to Tar. Bi et al. modeled the Tar sequence onto the Tsr coordinates, but retained the Af1503 HAMP domain in their model. Again, I question the biological relevance of that model. (It wasn't clear from the supplemental methods what Tar residues were included and what structural constraints were applied to the C-termini of the model.)

(3) The MD simulation of the Tar periplasmic domain provides valuable insight into how NaCl might trigger an attractant through this receptor. (That model doesn't contain any thermophilic parts and the simulations were carried out at a more biologically relevant 27°C.)

(4) The authors conclude (p.17) that "the five-residue region...connecting AS2 and MH1 is crucial for the [sign of the] signal output". However, I couldn't determine from the description of the Tar mutants made in that region ("random linkers") what their average number of amino acid changes was. Do multiple residues in this region need to be changed to get response inversion? The three inverted

receptors all had a proline at residue R269. Is that change alone sufficient for inversion? Without knowing how a Tar-R269P receptor behaves, the five-residue conclusion doesn't seem warranted.

figure comments and suggestions:

(1) Relevant MD simulations were reported in a Ph.D. thesis by P. Orekhov (download at: <https://repositorium.uni-osnabrueck.de/handle/urn:nbn:de:gbv:700-2016081014821>).

(2) Figures 1, 3, 4, 5: The legends should contain more detail about the experiments, for example, the strains used and the Tar expression levels from the plasmids. Data points on the Hill fits report mean and standard error for "three independent replicates" (Fig. 1) or "three replicates" (Fig. 3). Were the replicate experiments of Fig. 3 also independent? Does independence mean different cell growth cultures? What are the Hill coefficients for the various responses? The inverted responses seem to be less cooperative than that of the wild-type. Is this correct? If so, does the difference have mechanistic implications?

(3) Figures 3, 4, 5 and supplemental ones: It would be easier for readers to compare the sizes of the FRET changes if the YFP/CFP data were plotted at the same scale.

(4) Fig. 5: Should add "NaCl" in each panel.

comments on the text:

Overall, the presentation reads well; clear and engaging.

(1) p.3: "glutamate residue" should probably be changed to "glutamyl residue".

(2) Throughout: I suggest replacing "upward/downward" terminology with "outward/inward" terminology for the piston displacements.

(3) p.4: Perhaps change "four-helical" to "four-helix".

Reviewer #2 (Remarks to the Author):

Review of Bi et al. Nature Communications

This manuscript utilizes a well-characterized transmembrane chemoreceptor from *Escherichia coli* to identify mutational changes that invert from attractant to repellent the sensory response to compounds recognized in the periplasmic domain ligand-binding site. Inversion of chemotactic responses by mutational changes in chemoreceptors has been observed multiple times in the last 40 years of research on mechanisms chemotaxis by *E. coli* and other bacteria. Inversions of responses to chemical and physical (temperature and osmotic pressure) stimuli have been observed and potential mechanisms described. The manuscript cites five of these (references 34, 36, 60, 61 and 62).

Additional relevant publications include (Muskavitch MA et al. 1978 *Science* 201:63-5; Mizuno and Imae, 1984. *J. Bacteriol.* 159:360-367; Nishiyama et al. 1997. *J. Bacteriol.* 179:6573-6580; Umemura et al. 2002. *J. Bacteriol.* 277:1593-1598). In the context of adding to the list of mutationally induced inversions, new contributions made by Bi et al. are to 1) identify and characterize mutations that are inverted the response to an attractant recognized by a defined ligand-binding site, 2) obtain those mutations in two specific regions of the rod-link chemoreceptors structure, the transmembrane domain and interface of the HAMP domain and the extended four-helix coiled coil and 3) provide tantalizing suggestions of the mechanisms underlying the signal inversions by the use of molecular

modeling and molecular dynamics. The intertwined utilization of mutational analysis and molecular dynamics is a strength of the study. In addition, the work utilizes the two loci of signal inversion to map the sections of the rod-like chemoreceptor structure that appear to contain the element(s) responsive to several different compounds that act as chemo-repellents. This application is a clever use of their identification of the two inversion loci. The results provide new information about the way in which chemo-repellents act on chemoreceptors, a subject on which information is sparse, in part because it appears that many if not most repellents are recognized in ways other than binding at a conventional, stereospecific ligand-binding sites.

In the context of considering the appropriateness of this manuscript for Nature Communications, its strengths have been summarized in the previous paragraph. Major concerns about the manuscript are as follows. 1) Chemotactic signal inversion has often been viewed as an interesting curiosity, not a major contributor to fundamental understanding of chemoreceptor signaling. In part this is because the picture emerging of the chemoreceptor is of a structure delicately balanced among conformational alternatives in a way that many alterations can shift the balance and thus the output. Signal inversion is one of many examples and inversion to some stimuli can be induced by simply shifting the signaling state by changing the extent of adaptational modification, as shown for thermotaxis (see references listed above). Thus, the addition to the list of the ways to shift the balance by mutational changes that, in this case, invert the response to an attractant bound at the periplasmic ligand-binding site, seems of modest interest. 2) Identification of mutational changes that invert response to attractants bound at the specific, periplasmic domain would not be considered by the field to be qualitatively different than identifying changes that invert responses to physical stimuli like temperature or osmotic pressure. This is because it seems that both kinds of stimuli act on the same chemoreceptor equilibrium between a kinase-on and kinase-off conformation. Thus the particular way in which that equilibrium is shifted is not a major issue. 3) The modeling/molecular dynamics, which in combination with mutational analysis is a positive aspect of the work, is only valid if the artificial use of an arbitrary piece of the entire chemoreceptor provides information that reflects what would occur in the intact molecule. There is no compelling reason to accept that this is the case. It is unfortunate that this issue is not addressed in the manuscript. Yet it is crucial to the validity of the simulation data and thus the creditability of many of the most interesting conclusions. 4) The result and conclusions described in the manuscript would likely be of only modest interest to those in the field of bacterial chemotaxis and signaling. This is in part because of the concerns outlined above. But it is also because, the major conclusions in large part represent only additional evidence to the growing body of suggestive observations that support the currently attractive thinking about signaling within the receptor involving competition at borders between different helical registers or between more or less helical nature. Thus the results and conclusions of this work seem most likely to have some but not major influence on subsequent research in the field of bacterial chemotaxis.

Specific comments:

1. Of significant concern are the results shown for the microfluidic assay of chemotaxis and the conclusions drawn from them. The images shown in Figs. 1 and 3 are not convincing. They need sufficient explanation so that an interested and critical reader can know what would be seen for positive and for negative taxis as well as no response at all. The essence of the explanation needs to be provided in the text; the details can be in Materials and Methods and/or in the figure legends. As currently presented, without this information it appears to the reader that no taxis is documented since in the image of cells in a gradient it appears that none are visible.

2. Lines 22-23. It is an overstatement to write "Similarly unclear is how easily sensory properties of receptors are modified in the process of evolutionary adaptation to novel environments." The taxonomic diversity of ligand recognition, signaling polarity and modular organization seems

convincing evidence that modification is “easy” on the evolutionary time scale.

3. Lines 24, 104 and 308. In light of the multiple times inversions of chemotactic responses have been documented in the literature, the phrase “...we demonstrate surprising plasticity of bacterial chemoreceptors...” in the Abstract and its similar statements in the Introduction and Discussion seem an overstatement. More accurate would be a phrasing such as: “we provide additional examples of the plasticity of bacterial chemoreceptors...”.

4. Lines 78-80. It seems appropriate to add references to early documentation of the responses listed in these lines. For amino acids (Mesibov and Adler, 1972 *J Bacteriol.* 112:315-26), sugars (Adler et al. 1973 *J Bacteriol.* 115:824-47), metal ions, aromatic compounds and pH (Tso and Adler, 1974 *J. Bacteriol.* 118:560-576), pH (Repaske and Adler 1981 *J Bacteriol.* 145:1196-208), temperature (Maeda and Imae 1976 *Proc. Natl. Acad. Sci.* 76:91-95), and osmotic pressure (Li et al. 1988 *Proc Natl Acad Sci U S A.* 1988 85:9451-5).

5. Line 83. Add an early reference for localization of sensing regions (Krikos et al. 1985 *Proc. Natl. Acad. Sci.* 82: 1326–1330).

6. Line 122. It is important to include an explanation of how the level of expression relates to natural expression from the chromosomal gene (and provide detail in Materials and Methods).

7. Lines 158-160. It is very difficult to compare the structures shown in Fig. 2a. The authors need to find a better way to highlight the differences.

8. Line 171. Why refer to this region as the “conjunction region” when most of publications refer to it as the “control cable”?

9. Line 173. It would be more precise to substitute “in the same direction” for “similar”.

10. Paragraph beginning on line 180. The text needs to be specific about which bending angle is being discussed, since Table S1 shows that the two helices (from the occupied and unoccupied subunit) have different bending angles and that the greater difference from receptor with ligand bound and without is in the unoccupied subunit. A similar pattern appears to be shown in Table S3. Clear explanations are necessary.

11. Line 285. Add reference to Li et al. 1988 *Proc Natl Acad Sci U S A.* 1988 85:9451-5.

12. Table 1. Given the standard deviations of the values presented, the values themselves have too many significant figures.

13. Figure 1. The legend refers to error bars on the plots of normalized response versus concentration, but no error bars are visible.

14. Figure S4. The colors for the two states in S4b are very hard to distinguish. More difference between the colors would help.

15. Figure S4 legend. Line 47 there is a superscript “1” after (interface C), but no footnote corresponding to that number.

16. Table S1. There needs to be an explanation of the significance of stimulation indexes A, B and C for each genetic variant.

17. Table S1. The prime on TM2'a is very difficult to see.

Point-by-point response to reviewers' comments

We would like to thank both reviewers for their thorough and critical review of the manuscript and for their overall positive appreciation of our work as well as critical comments and suggestions for improvement of the manuscript. We have revised the manuscript according to the reviewers' suggestions, particularly improving the presentation and discussion of computer simulations and also discussing more extensively the significance of our work in the context of previously published results. Below are our detailed responses to individual reviewers' comments.

Reviewer #1

"In this manuscript, Bi et al., report characterization of mutant aspartate chemoreceptors that show inverted kinase control responses to various stimuli. Two types of inverted mutants were studied: (1) Receptors with 2-3 additional residues in the membrane-spanning TM2 helices that couple conformational changes in the ligand-binding domain to those in the cytoplasmic HAMP domain. (2) Receptors with randomized five-residue sequences near the junction of the HAMP domain helices and the methylation helices that control kinase output and sensory adaptation. The mutant receptors were cleverly used to determine what part of the molecule was likely responsible for sensing various less well understood stimulus inputs. The mutant receptor behaviors are quite interesting and shed new light on transmembrane and intracellular signaling by this well-studied family of bacterial chemoreceptors.

Molecular dynamics simulations of portions of the mutant receptors provided mechanistic explanations for the signal inversion. However, I have concerns about the validity of the structural models used for most of the MD simulations. I do not feel that those simulations make a convincing case for the mechanisms proposed. I offer a

number of suggestions for improving the solidity of the scientific case and the clarity of the presentation.

We thank the Reviewer for the suggestions on improvement of our MD simulations and their presentation in the manuscript.

major issues:

(1) The model for the MD simulations of the periplasmic domain-TM bundle-HAMP bundle was essentially the one used by Park et al (2011) in an earlier MD study. It employs a chimeric protein in which part of the periplasmic domain and the TM bundle of Tar are joined to a HAMP domain (Af1503) from a hyperthermophile. From the supplementary methods, it appears that the simulations were then carried out at 50°C. Thus, part of the model protein evolved to operate 15-20 degrees below the simulation temperature and part evolved to operate at 20-30 degrees above the simulation temperature. Given the conflicting structural and dynamic interactions between those two disparate parts, it's difficult to see how that model could reliably reproduce what happens in the native receptor at its normal operating temperature. Some of the constraints imposed on the models of the mutant proteins (e.g., wild-type piston displacements and maintenance of TM2 helicity) might also have influenced the simulation outcomes. (It was unclear exactly what structural constraints, if any, were placed at the C-termini of the HAMP helices.)"

Here, as mentioned by the referee, we relied on the previously published work (Park et al., 2011; Ref. 16), and because of that we did not provide further justification for using this chimeric structure. This is corrected in the revised version of the manuscript, where we now discuss this simulated hybrid structure and justify its use for MD simulations in the Results and Methods sections. One key point in this respect is that the hybrid that carries the Af1503 HAMP domain instead of the native HAMP domain

of Tar was already experimentally shown to be functional (transmit chemotactic signals) in Hulko et al., 2006 (Ref. 13). Moreover, results of previous simulations performed by Park et al are consistent with several experimental results obtained for native Tar. Both of these argue in favour of the biological relevance of this model. As there is no experimentally determined structure of the native HAMP domain of Tar and the quality of the homology modelling for the Tar HAMP domain is not sufficiently high for MD simulations, because of the low sequence identity between the structures of *E. coli* and *A. fulgidus*, we think using structure of this (functional) hybrid for MD simulations is indeed the best available option. But we fully agree that these arguments need to be explicitly made in the manuscript, as done now.

As for performing the simulations at 50°C, here we again followed the previous work of Park et al (Ref. 16). Notably, this relatively high temperature is required for simulations of the DPPC lipid used in the model. As mentioned in the revised version of the manuscript, there is a body of literature besides Park et al where simulations in DPPC and at 50°C were used to simulate dynamics of e.g. *E. coli* or human mesophilic membrane proteins (Refs. 72-74). Nevertheless, to further address the Reviewer's point, we performed additional simulations at 27°C in another type of lipid, POPC (Refs. 30 and 75). To simultaneously address the question #3) of Reviewer #2, these simulations were done for a longer receptor fragment containing methylation helix bundle. The same principal differences between the stimulus-induced conformational changes in wild-type Tar vs Tar^{TM2+2I} were observed in both simulations, confirming that our results are not critically influenced by the type of lipids, the simulation temperature, or the length of receptor fragment. These new simulations are shown in SI (Fig. S7, Tables S2 and S4) and discussed in the main text (pages 10-12) and in SI Methods.

We further specified all the used structural restraints in Tables S10, S11 and in SI Methods. To maintain the overall structure of TM helices, the helix restraints were initially applied in implicit solvent simulations to build the models of the wild-type and mutant receptors and then decreased gradually to zero in explicit solvent

simulations within 60 ns. During the final simulations, no helices restraints were applied any more. No restraints were placed at the C-termini of the HAMP domain.

"(2) The model for MD simulations of the HAMP-methylation helix interactions was based on the crystal structure of a chimeric protein containing the Af1503 HAMP domain joined to the signaling domain of Tsr, an E. coli receptor closely related to Tar. Bi et al. modeled the Tar sequence onto the Tsr coordinates, but retained the Af1503 HAMP domain in their model. Again, I question the biological relevance of that model. (It wasn't clear from the supplemental methods what Tar residues were included and what structural constraints were applied to the C-termini of the model.)

For the reasons described above in our reply to point #1, we believe that the use of the Tar-Af1503 hybrid structure is well justified and biologically relevant. We have now further expanded the description of the model in SI, also specifying and justifying used structural restraints in Tables S10, S11 and in SI Methods. We apologize that it was not fully explained in the original version of the manuscript. The position restraints in x - y plane parallel to membrane were applied to the C-termini of the models during simulations to maintain the helical structure at C-termini. These atoms are still free in z direction (vertical to membrane) to make inward or outward movement possible. We think these restraints have little influence on the dynamics of the region that we are interested in, as the restrained atoms of C-termini are far away from the region of interest.

We additionally validated our results by performing simulations of the same structure at 27°C in POPC (see our reply to point #1) for the wild-type Tar and Tar^{HAMP_GVPQM}. These results are now shown in SI (Fig. S13, Tables S6 and S7).

(3) The MD simulation of the Tar periplasmic domain provides valuable insight into

how NaCl might trigger an attractant through this receptor. (That model doesn't contain any thermophilic parts and the simulations were carried out at a more biologically relevant 27 °C.)"

We agree that this simulation provides valuable insight.

"(4) The authors conclude (p.17) that "the five-residue region...connecting AS2 and MHI is crucial for the [sign of the] signal output". However, I couldn't determine from the description of the Tar mutants made in that region ("random linkers") what their average number of amino acid changes was. Do multiple residues in this region need to be changed to get response inversion? The three inverted receptors all had a proline at residue R269. Is that change alone sufficient for inversion? Without knowing how a Tar-R269P receptor behaves, the five-residue conclusion doesn't seem warranted."

Although by stating that this “five-residue region... is crucial” we did not mean to claim that all five residues are necessarily crucial, we thank the Reviewer for suggesting to explore the minimal requirement for sign inversion in this case. We have now performed additional experiments, showing that mutant carrying R269P substitution alone is unresponsive to MeAsp, likely because it is locked in inactive state. However, an additional exchange of G271 to the hydrophobic residue Y or M, as found in all three functional mutants, restored the activity and indeed was sufficient to mediate repellent response to MeAsp. These new results (shown in Fig. S14) thus suggest that replacements of just two residues in this region are sufficient for response inversion, enabling us to make a more precise statement on pages 14, 15 and 20.

"figure comments and suggestions:

(1) Relevant MD simulations were reported in a Ph.D. thesis by P. Orekhov (downloadat:<https://repositorium.uni-osnabrueck.de/handle/urn:nbn:de:gbv:700->

2016081014821)."

We thank the Reviewer for pointing this out; this Ph.D. thesis is now cited as Ref. 15 and a publication resulting from this thesis is cited as Ref. 30.

"(2) Figures 1, 3, 4, 5: The legends should contain more detail about the experiments, for example, the strains used and the Tar expression levels from the plasmids. Data points on the Hill fits report mean and standard error for "three independent replicates" (Fig. 1) or "three replicates" (Fig. 3). Were the replicate experiments of Fig. 3 also independent? Does independence mean different cell growth cultures? What are the Hill coefficients for the various responses? The inverted responses seem to be less cooperative than that of the wild-type. Is this correct? If so, does the difference have mechanistic implications?"

We have added more details to the figure legend, as suggested by the Reviewer. In the previous version of the manuscript, the strains used and the induction levels of Tar were described in the methods and in Table S4 (now Table S8 in the revised manuscript). The expression level of Tar under our experimental conditions was quantified in a previous study from our lab (Ref. 67). We now specify strains in the legends and receptor expression in the results and methods (pages 7 and 24). We also make it clear in the legends that error bars for all FRET measurements indeed reflect three independent biological replicates.

We now also compare the values of Hill coefficients for the responses of wild-type Tar and mutant receptors Tar^{TM2+2I} (pages 7, 8, Fig. 2) and Tar^{HAMP_GVPQM} (pages 12, 13, Fig. 3), and discuss possible significance of the observed differences. Both inverted responses were indeed less cooperative, and at least in the case of Tar^{TM2+2I} it might be related to less efficient clustering of this mutant (now shown as Fig. S2). In contrast, Tar^{HAMP_GVPQM} formed normal clusters (new Fig. S11), indicating that lower cooperativity might be related to differences in local receptor interactions within

clusters.

"(3) Figures 3, 4, 5 and supplemental ones: It would be easier for readers to compare the sizes of the FRET changes if the YFP/CFP data were plotted at the same scale."

Although we principally agree that plotting the YFP/CFP data at the same scale would facilitate comparisons of the response magnitudes, we believe that scaling according to the amplitude of signal makes it easier to visualize the sign of the response. Because the latter is more important in the context of our manuscript, we prefer to keep this scaling.

"(4) Fig. 5: Should add "NaCl" in each panel."

We now added "NaCl" in each panel as suggested.

"comments on the text:

Overall, the presentation reads well; clear and engaging.

(1) p.3: "glutamate residue" should probably be changed to "glutamyl residue".

(2) Throughout: I suggest replacing "upward/downward" terminology with "outward/inward" terminology for the piston displacements.

(3) p.4: Perhaps change "four-helical" to "four-helix"."

We thank the Reviewer for finding our presentation clear and engaging. All of the suggested corrections have been made.

Reviewer #2

"This manuscript utilizes a well-characterized transmembrane chemoreceptor from Escherichia coli to identify mutational changes that invert from attractant to repellent the sensory response to compounds recognized in the periplasmic domain ligand-binding site. Inversion of chemotactic responses by mutational changes in chemoreceptors has been observed multiple times in the last 40 years of research on mechanisms chemotaxis by E. coli and other bacteria. Inversions of responses to chemical and physical (temperature and osmotic pressure) stimuli have been observed and potential mechanisms described. The manuscript cites five of these (references 34, 36, 60, 61 and 62). Additional relevant publications include (Muskavitch MA et al. 1978 Science 201:63-5; Mizuno and Imae, 1984. J. Bacteriol. 159:360-367; Nishiyama et al. 1997. J. Bacteriol. 179:6573-6580; Umemura et al. 2002. J. Bacteriol. 277:1593-1598). In the context of adding to the list of mutationally induced inversions, new contributions made by Bi et al. are to 1) identify and characterize mutations that are inverted the response to an attractant recognized by a defined ligand-binding site, 2) obtain those mutations in two specific regions of the rod-link chemoreceptors structure, the transmembrane domain and interface of the HAMP domain and the extended four-helix coiled coil and 3) provide tantalizing suggestions of the mechanisms underlying the signal inversions by the use of molecular modeling and molecular dynamics. The intertwined utilization of mutational analysis and molecular dynamics is a strength of the study. In addition, the work utilizes the two loci of signal inversion to map the sections of the rod-like chemoreceptor structure that appear to contain the element(s) responsive to several different compounds that act as chemo-repellents. This application is a clever use of their identification of the two inversion loci. The results provide new information about the way in which chemo-repellents act on chemoreceptors, a subject on which information is sparse, in part because it appears that many if not most repellents are recognized in ways other than binding at a conventional, stereospecific ligand-binding sites.

We thank the Reviewer for this clear summary of the main novel findings made in our manuscript, and for acknowledging strengths of our work. Regarding additional publications mentioned as relevant by the Reviewer, two refer to the same inversion of temperature response as Ref. 60 that were already cited in the original version of the manuscript (but we have nevertheless added these additional references to the revised version, Refs. 61 and 62), and the third reference is the same as Ref. 40 that was already included in the previous version of the manuscript. And although the remaining publication, Muskavitch et al., 1978, indeed uses the terminology of “inversion”, from what we know now (and the authors could not know in 1978) the mutations it reported did not actually invert but simply abolished response mediated by one of the chemotaxis receptors, so that the opposite response mediated by another receptor became apparent.

In the context of considering the appropriateness of this manuscript for Nature Communications, its strengths have been summarized in the previous paragraph. Major concerns about the manuscript are as follows.

1) Chemotactic signal inversion has often been viewed as an interesting curiosity, not a major contributor to fundamental understanding of chemoreceptor signaling. In part this is because the picture emerging of the chemoreceptor is of a structure delicately balanced among conformational alternatives in a way that many alterations can shift the balance and thus the output. Signal inversion is one of many examples and inversion to some stimuli can be induced by simply shifting the signaling state by changing the extent of adaptational modification, as shown for thermotaxis (see references listed above). Thus, the addition to the list of the ways to shift the balance by mutational changes that, in this case, invert the response to an attractant bound at the periplasmic ligand-binding site, seems of modest interest.

We would argue that the reason why previous reports of mutations inverting the responses to non-canonical stimuli, such as pH or temperature, remained “an

interesting curiosity” is exactly because molecular mechanism of signalling by these non-canonical stimuli has not been elucidated. Therefore the interpretation and the meaning of the observed response inversion remained unclear for over three decades since their first observation. Moreover, the mutations that inverted these responses did not invert the canonical ligand response. These observations therefore failed to provide any useful information about receptor signalling. In the specific case of temperature, mentioned by the Reviewer, it is currently still unknown how temperature elicits the response, let alone how this response is inverted by modifying receptor adaptation sites.

Moreover, although many mutations are known to shift the equilibrium signalling state of the receptors to being more or less active, inverting a sign of the response is qualitatively different. So far, not a single mutant has been reported that inverts signalling to canonical ligands, whereas hundreds of mutants shifting the on-off equilibrium have been described.

In our opinion, a fundamental difference between these previous studies and our work is that we could identify such mutants for canonical ligands and finally provide mechanistic interpretations for the observed inversion, thus for the first time turning inversion mutants from an interesting curiosity into tools to better understand receptor signalling.

2) Identification of mutational changes that invert response to attractants bound at the specific, periplasmic domain would not be considered by the field to be qualitatively different than identifying changes that invert responses to physical stimuli like temperature or osmotic pressure. This is because it seems that both kinds of stimuli act on the same chemoreceptor equilibrium between a kinase-on and kinase-off conformation. Thus the particular way in which that equilibrium is shifted is not a major issue.

See our response to point #1 above. We believe that inverting the sign of the response is fundamentally different from simply shifting the equilibrium. Moreover, physical

stimuli such as osmotic pressure or temperature seem to act downstream of the sensory domain (results in this manuscript suggest this for osmolarity, and our unpublished work for temperature), thus likely shifting receptor equilibrium in a very different way than canonical ligands.

3) The modeling/molecular dynamics, which in combination with mutational analysis is a positive aspect of the work, is only valid if the artificial use of an arbitrary piece of the entire chemoreceptor provides information that reflects what would occur in the intact molecule. There is no compelling reason to accept that this is the case. It is unfortunate that this issue is not addressed in the manuscript. Yet it is crucial to the validity of the simulation data and thus the creditability of many of the most interesting conclusions.

Although we in principle understand the concern of the reviewer, simulating only the relevant part of the protein is a common practice in MD simulations of modular multi-domain proteins, including previous studies for bacterial chemoreceptors using all-atom models (incl. Refs. 10, 16, 52 and 73). There is abundant evidence that such simulations are valid, providing mechanistic insights and agreeing with experiments. Although we could in principle simulate the entire structure of the receptor, in our opinion these simulations are not necessary, as simulating parts of the receptors that are far remote from the region of interest is unlikely to provide us with much useful information. At the same time, such simulations would be extremely expensive in terms of the computational time, limiting our ability to generate statistics and explore effects of different mutations as done in the current study. We now discuss in the text the physiological relevance of simulating shorter receptor fragments.

However, to additionally address this point we also performed additional simulations for wild-type Tar and Tar^{TM2+21} using a longer receptor fragment also containing methylation helix bundle (Fig. S7, Tables S2 and S4, SI Methods). As expected, there was no difference in the stimulus-induced conformational changes of wild-type Tar

and Tar^{TM2+2I} when compared to the original simulations using a shorter receptor fragment, strongly arguing that the short fragment already contains all the necessary structural determinants. This is now discussed in the main text (pages 10-12).

4) The result and conclusions described in the manuscript would likely be of only modest interest to those in the field of bacterial chemotaxis and signaling. This is in part because of the concerns outlined above. But it is also because, the major conclusions in large part represent only additional evidence to the growing body of suggestive observations that support the currently attractive thinking about signaling within the receptor involving competition at borders between different helical registers or between more or less helical nature. Thus the results and conclusions of this work seem most likely to have some but not major influence on subsequent research in the field of bacterial chemotaxis."

See our responses above. Although our results are indeed consistent with the general model of receptor signalling that involves competition between conformations of individual domains, they help to understand at the structural level how this competition occurs, i.e. how the signal is transmitted from one domain to the other, which is quite important for understanding signal transduction through receptors.

"Specific comments:

1. Of significant concern are the results shown for the microfluidic assay of chemotaxis and the conclusions drawn from them. The images shown in Figs. 1 and 3 are not convincing. They need sufficient explanation so that an interested and critical reader can know what would be seen for positive and for negative taxis as well as no response at all. The essence of the explanation needs to be provided in the text; the details can be in Materials and Methods and/or in the figure legends. As currently presented, without this information it appears to the reader that no taxis is documented since in the image of cells in a gradient it appears that none are visible."

We apologize for not sufficiently clarifying the microfluidic experiments. Although the microfluidic device used in the study was already described before in Ref. 69, we agree that describing the assay in this manuscript is nevertheless important. We now expanded the corresponding description on pages 8 and 25, Fig. S3, and legends of Fig. 1 and Fig. 3.

"2. Lines 22-23. It is an overstatement to write "Similarly unclear is how easily sensory properties of receptors are modified in the process of evolutionary adaptation to novel environments." The taxonomic diversity of ligand recognition, signaling polarity and modular organization seems convincing evidence that modification is "easy" on the evolutionary time scale.

Although we agree that there is plenty of evidence that receptor properties can change under selection on the evolutionary time scale (millions of years), it is by no means clear how rapidly these changes can occur. We now emphasize more clearly this difference.

3. Lines 24, 104 and 308. In light of the multiple times inversions of chemotactic responses have been documented in the literature, the phrase "...we demonstrate surprising plasticity of bacterial chemoreceptors..." in the Abstract and it similar statements in the Introduction and Discussion seem an overstatement. More accurate would be a phrasing such as: "we provide additional examples of the plasticity of bacterial chemoreceptors..."

As mentioned above (points #1, 2 and 4), we believe that there is a fundamental difference between these different types of inversion. However, we modified the

abstract to improve the phrasing.

"4. Lines 78-80. It seems appropriate to add references to early documentation of the responses listed in these lines. For amino acids (Mesibov and Adler, 1972 J Bacteriol. 112:315-26), sugars (Adler et al. 1973 J Bacteriol. 115:824-47), metal ions, aromatic compounds and pH (Tso and Adler, 1974 J. Bacteriol. 118:560-576), pH (Repaske and Adler 1981 J Bacteriol. 145:1196-208), temperature (Maeda and Imae 1976 Proc. Natl. Acad. Sci. 76:91-95), and osmotic pressure (Li et al. 1988 Proc Natl Acad Sci U S A. 1988 85:9451-5).

5. Line 83. Add an early reference for localization of sensing regions (Krikos et al. 1985 Proc. Natl. Acad. Sci. 82: 1326–1330)."

We now cited these additional references.

"6. Line 122. It is important to include an explanation of how the level of expression relates to natural expression from the chromosomal gene (and provide detail in Materials and Methods)."

More details on the expression level of Tar are now provided in the manuscript (pages 7 and 24); it was quantified in our previous study (Ref. 67).

7. Lines 158-160. It is very difficult to compare the structures shown in Fig. 2a. The authors need to find a better way to highlight the differences.

We did the structural alignment to better highlight the difference between the structures in Fig. S6.

8. Line 171. Why refer to this region as the "conjunction region" when most of

publications refer to it as the “control cable”?

In Fig. 2c we calculated the average helicity of residues ²¹¹GIRRMLLT²¹⁸ connecting TM2 and AS1 of the HAMP domain. These eight residues include the control cable GIRRM and the first three residues LLT of AS1 (Fig. 1a). We clarified this in the revised manuscript (page 10).

"9. Line 173. It would be more precise to substitute “in the same direction” for “similar”."

We revised this sentence according to the Reviewer's suggestion.

"10. Paragraph beginning on line 180. The text needs to be specific about which bending angle is being discussed, since Table S1 shows that the two helices (from the occupied and unoccupied subunit) have different bending angles and that the greater difference from receptor with ligand bound and without is in the unoccupied subunit. A similar pattern appears to be shown in Table S3. Clear explanations are necessary."

We thank the Reviewer for pointing it out. We clarified the description in the revised text (pages 11,12) and Tables S1-S4.

"11. Line 285. Add reference to Li et al. 1988 Proc Natl Acad Sci U S A. 1988 85:9451-5."

We now cited this reference as Ref. 45.

"12. Table 1. Given the standard deviations of the values presented, the values themselves have too many significant figures."

We reduced the number of significant figures for the values.

"13. Figure 1. The legend refers to error bars on the plots of normalized response versus concentration, but no error bars are visible."

Some error bars are smaller than the symbol size so they are invisible in the plots. We now explained this in the figure legends.

"14. Figure S4. The colors for the two states in S4b are very hard to distinguish. More difference between the colors would help."

We now changed the color for the two states in Fig. S6.

"15. Figure S4 legend. Line 47 there is a superscript "1" after (interface C), but no footnote corresponding to that number."

In the legend of Fig. S8, we clarified that the superscript "1" refers to the supplementary reference 1.

"16. Table S1. There needs to be an explanation of the significance of simulation indexes A, B and C for each genetic variant."

"17. Table S1. The prime on TM2'a is very difficult to see."

In the revised Tables S1-S4 we clarified that the simulation indexes A, B and C for each genetic variant are three independent MD simulation trajectories. We also enlarged the primes for better visibility.

Reviewers' comments:

Reviewer #3 (replaces referee #1):

Review of Inverted signaling by bacterial chemotaxis receptors as submitted for publication in Nature Communications.

The authors experimentally identify a number of interesting mutations in two key areas of the bacterial chemoreceptor Tar, which give rise to an inverted signaling phenotype. Molecular dynamics (MD) simulations are used to investigate characteristics of the structural and dynamical changes caused by these mutations to provide molecular insight into the mechanism of transmembrane and HAMP signaling in Tar. Based on the location of mutations causing signaling inversion, the authors assign the probable sensation of various chemoeffectors (especially certain enigmatic repellents) to localized regions of Tar.

Regarding the experimental aspects of the present study, reviewers 1 and 2 have more than adequately assessed the quality and significance of the presented results. To these concerns, I've nothing more to add. However, I must echo their concerns surrounding certain computational aspects of the study (even in light of the revised manuscript and additional simulations). While it is clear the authors have experience in setting up and conducting MD simulations; the subsequent trajectory analysis is poorly described and, in my opinion, not strong enough to provide adequate support for several of the conclusions drawn from it. In addition, the presented results do not seem to provide much in the way of new understanding or predictive power.

A detailed discussion of each issue follows.

1) Major issue: The authors do not demonstrate before beginning production simulations that any of their models have, in fact, equilibrated to the various structural perturbations introduced. In particular, they do not report monitoring any quantities (e.g., basic RMSD or area per lipid) to help decide when the model has relaxed into the new local minimum created by altering the structure. According to the discussion in the SI and Table S10, it seems that a single protocol for all models was used regardless of where the structure was altered, the extent of modeling required to bring about the change, or the uniqueness of any particular MD run. This is especially worrisome for structural changes made within the membrane bilayer, which relaxes on a timescale of hundreds-of-nanoseconds---much longer than the equilibration times allowed here. In addition, no time traces for any of the analyzed quantities are provided. As such, it is not clear whether certain of the observed differences (e.g., changes in helical propensity or the relative distance between two residues) are the result of the transduction of structural or dynamical information as proposed, or rather the consequence of the different models responding similarly (if the same mutation) or dissimilarly (if a different mutation) to a significant perturbation in structure (even when beginning from randomized initial velocities). This is simply not acceptable in a study such as this one whose major conclusions depend significantly on MD results.

2) For control cable helicity, the reported trend is interesting and in line with ideas previously proposed in the literature. However, the authors do not address at all how these changes in helicity are manifest. How does the change in helicity vary between the ligand-bound and ligand-unbound monomers in the AH models? Which specific residues are involved? Is there a complete helix-to-coil transition or are a few hydrogen bonds broken here and there? Is helicity ever regained once lost? What role do lipid/protein interactions play? The authors could have addressed such questions through the per-residue secondary-structure time traces for each monomer.

3) Regarding TM2 bending angle and control cable longitudinal position, again the time traces of these variables should have been given. More importantly, however, it's not demonstrated or apparent that the chosen reaction coordinates properly monitor the quantities ascribed to them. Why is TM2 bending angle measured in such an obscure way? More convincing analysis would measure and compare the local curvature of the entire TM2/control-cable/AS1 region over time. In addition, it's not clear how the relative z-hat distance between a single residue (I220) can be used as a proxy for the longitudinal displacement of the whole control cable AS1 helix, especially as portions of this segment are in different solvent environments. It is further proposed that this displacement may give rise to repacking of the HAMP. It is not clear, though, if the authors actually looked for any such repacking or not.

4) It's concerning that the authors report the breaking of the TM2 helix in the TM2+21 AH model near the middle of the membrane bilayer. Such a break requires significant energy in a properly equilibrated and tightly packed 4-helix transmembrane bundle. Did this happen in all of the simulations of this model? When in the simulations did the event take place?

5) Major issue: The whole of the conclusions drawn by the authors to explain the effects of the AS2/MH1 linker mutations for signaling inversion are dependent on RMSF measurements. While RMSF is a widely used pseudo-metric for describing protein fluctuations, it is heavily dependent on the particulars of the alignment and best suited for globular proteins. In particular, when dealing with long, coiled-coil bundles such as Tar, the natural, global bending of the bundle makes it difficult to align these structures in a way appropriate to allow for direct comparison of local RMSFs. No detailed description of the alignment process is given, rendering the present comparisons difficult to interpret. The lack of protocol detail here is especially concerning given the shape of the provided RMSF traces in Figure S12, which show nearly identical shapes differentiated by global shifts up or down. This implies that the changes in fluctuations are not internal to the domains themselves, but due to some sort of global fluctuation possibly arising from misalignment. This severely undermines the offered piston-force/support-force explanation. A more convincing and interesting analysis would have involved the quantification of changes in the correlated motions between individual residues in the various regions of the HAMP/linker/MH bundle.

6) The simulations of the Tar periplasmic domain in two concentrations of NaCl appear to have been "thrown in" as an afterthought. The simulation analysis is not clearly described, nor is the robustness of the obtained results demonstrated. It is not clear how the two mentioned changes (inter-monomer rotation and sliding between helix a1 and a4) are actually "similar to the conformation triggered by the binding of aspartate," or what other changes are present in the structure. Though the experiments and logic used for mapping sensory regions in Tar are intriguing, considering the sparsity of MD-related detail, it's difficult to tell whether the presented simulations support the authors' conclusions or not.

In conclusion, the inverted signaling mutants presented are interesting experimentally as highlighted by the previous reviewers. In addition, the use of MD simulation in principle provides a solid framework to investigate the types of molecular questions proposed here, and the authors have proposed a rigorous set of MD simulations to investigate them. However, for the reasons cited above, the poor description of the equilibration process and analyses raise many questions that cannot be overlooked. More importantly, however, the authors miss numerous opportunities to provide new insight into transmembrane and HAMP signaling by largely limiting their analyses to those quantities already specified by Park et. al. (2011) and mechanistic concepts already proposed in the literature. As such, the MD results provide little new understanding or predictive power, and I unfortunately cannot recommend the article for publication in Nature Communications.

Reviewer #4 (replaces referee #2):

My role as an additional reviewer was to comment on the revised manuscript and to assess the authors' response to reviewer #2 concerns.

Overall, this is an interesting paper with a large amount of experimental and simulation data. In my opinion, results do justify conclusions, especially after all the changes in response to reviewers' comments were made. Reviewers of the original submission did an excellent job in identifying potential issues and I have no specific concerns of my own.

Authors responded to reviewers' concerns adequately and additional data was provided where needed.

The main issue raised by Reviewer #2 was the impact of this study, which was viewed as modest. The authors, on the other hand, argued that their work provides mechanistic explanations for a phenomenon for the first time.

I agree with the authors that their work explains the mechanism of "the inverted response". Indeed, because of their work, this phenomenon can be understood in molecular terms.

However, I also agree with Reviewer #2. Attempts to present this study as something bigger than that (e.g. plasticity, evolution) are not necessarily convincing. The fact that single mutations can dramatically alter protein function is well known, e.g. aberrant signaling by protein kinases leading to cancers.

To sum up, I do think that this paper will have impact on the chemotaxis field. Whether or not this impact is big enough to justify publication in Nature Communications is the Editor's call.

Point-by-point response to reviewers' comments

We would like to thank both new reviewers for their mostly positive comments on our manuscript and also for specific suggestions on improvement of the analysis and discussion of MD simulation. We have revised the manuscript accordingly, addressing all points raised by the reviewers.

Reviewer #3 (replaces referee #1):

"Review of Inverted signaling by bacterial chemotaxis receptors as submitted for publication in Nature Communications.

The authors experimentally identify a number of interesting mutations in two key areas of the bacterial chemoreceptor Tar, which give rise to an inverted signaling phenotype. Molecular dynamics (MD) simulations are used to investigate characteristics of the structural and dynamical changes caused by these mutations to provide molecular insight into the mechanism of transmembrane and HAMP signaling in Tar. Based on the location of mutations causing signaling inversion, the authors assign the probable sensation of various chemoeffectors (especially certain enigmatic repellents) to localized regions of Tar.

Regarding the experimental aspects of the present study, reviewers 1 and 2 have more than adequately assessed the quality and significance of the presented results. To these concerns, I've nothing more to add. However, I must echo their concerns surrounding certain computational aspects of the study (even in light of the revised manuscript and additional simulations). While it is clear the authors have experience in setting up and conducting MD simulations; the subsequent trajectory analysis is poorly described and, in my opinion, not strong enough to provide adequate support

for several of the conclusions drawn from it. In addition, the presented results do not seem to provide much in the way of new understanding or predictive power.

Although we have already substantially expanded the description of MD simulations in the previous revision, we nevertheless fully agree with the referee that additional details of simulations should be provided and – even more importantly – that more information could be extracted from our MD results. We have now addressed both of these issues. Importantly, these additional analyses not only confirmed all of our previous conclusions but also indeed provided novel insights into receptor signaling, as proposed by Reviewer #3.

A detailed discussion of each issue follows.

1) Major issue: The authors do not demonstrate before beginning production simulations that any of their models have, in fact, equilibrated to the various structural perturbations introduced. In particular, they do not report monitoring any quantities (e.g., basic RMSD or area per lipid) to help decide when the model has relaxed into the new local minimum created by altering the structure. According to the discussion in the SI and Table S10, it seems that a single protocol for all models was used regardless of where the structure was altered, the extent of modeling required to bring about the change, or the uniqueness of any particular MD run. This is especially worrisome for structural changes made within the membrane bilayer, which relaxes on a timescale of hundreds-of-nanoseconds---much longer than the equilibration times allowed here. In addition, no time traces for any of the analyzed quantities are provided. As such, it is not clear whether certain of the observed differences (e.g., changes in helical propensity or the relative distance between two residues) are the result of the transduction of structural or dynamical information as proposed, or rather the consequence of the different models responding similarly (if the same mutation) or dissimilarly (if a different mutation) to a significant

perturbation in structure (even when beginning from randomized initial velocities). This is simply not acceptable in a study such as this one whose major conclusions depend significantly on MD results."

In the revised manuscript, we now provide the C α RMSD for the trajectories of the model-building simulations for the AA and AH states of wild-type Tar (Fig. S5a). The structures are well equilibrated before the production run, which shows the quality of the AA and AH model built for wild-type Tar. We also showed the C α RMSD from the initial structure as a function of simulation time for the three independent MD simulations for the AA and AH states of wild-type and mutant Tar receptors simulated in DPPC at 323 K or POPC at 300 K (Figs. S5b and S8). In these simulations, the AA and AH models undergo 100-200 ns equilibration and then mostly remain around 3 Å during the rest of simulation time, indicating that the trajectories can be considered relatively stable.

Our protocol to generate the models of mutants is based on the idea that the transmembrane domain should be α -helical. Although this protocol is used to generate model for different mutants, the models for a specific mutant generated by Modeller are very similar (with C α RMSD less than 1 Å). We selected three best models based on DOPE values (lowest energy) of Modeller and used them in our MD simulations. Based on the extended equilibration process (150 ns for wild-type Tar and 80 ns for each mutant model before production, Tables S8 and S9) and consistency of results of different MD simulations, as well as the RMSD values that indicate equilibration during production simulations, we think that our simulations provide robust results for both wild-type and mutant receptors.

"2) For control cable helicity, the reported trend is interesting and in line with ideas previously proposed in the literature. However, the authors do not address at all how these changes in helicity are manifest. How does the change in helicity vary between

the ligand-bound and ligand-unbound monomers in the AH models? Which specific residues are involved? Is there a complete helix-to-coil transition or are a few hydrogen bonds broken here and there? Is helicity ever regained once lost? What role do lipid/protein interactions play? The authors could have addressed such questions through the per-residue secondary-structure time traces for each monomer."

We thank the Reviewer for these suggestions. We now used the DSSP program to analyze the time-evolution of secondary structure for the residues ²¹¹GIRRMLLT²¹⁸ in the AA and AH states of the wild-type Tar and Tar^{TM2+2I} (Fig. S10). In the AH state, the ligand-occupied and ligand-free monomer were analyzed individually and the average helicity of the junction residues in the ligand-free monomer was also shown (Fig. S7 and Fig. S9c). We described the corresponding results in detail on pages 11-12. These results indeed show how helicity changes spread along the control cable, and we discuss them in the context of experimental literature.

We also explored the role of the protein-lipid interactions on the conformation and dynamics of the junction residues, as suggested by the Reviewer. We calculated the average electrostatic interactions as well as the hydrogen bonds formed between the residues ²¹⁰YGIRRMLLTP²¹⁹ and the phospholipids in the AA and AH states of wild-type Tar and Tar^{TM2+2I} (Fig. S13, pages 13-14). These simulations indeed show that the protein-lipid electrostatic interactions exhibit opposite changes in wild-type Tar and Tar^{TM2+2I}. In particular, the R213R214-lipid interactions appear to be important. The decrease in the electrostatic interactions and outward sliding of the control cable and AS1 of HAMP affects the conformation of these residues and decreases their structural dynamics, as shown in the RMSF analysis in Fig. S14, pages 14-15. These results suggest the importance of protein-lipid interactions in signal transduction, which is now discussed.

"3) Regarding TM2 bending angle and control cable longitudinal position, again the time traces of these variables should have been given. More importantly, however, it's

not demonstrated or apparent that the chosen reaction coordinates properly monitor the quantities ascribed to them. Why is TM2 bending angle measured in such an obscure way? More convincing analysis would measure and compare the local curvature of the entire TM2/control-cable/AS1 region over time. In addition, it's not clear how the relative z-hat distance between a single residue (I220) can be used as a proxy for the longitudinal displacement of the whole control cable AS1 helix, especially as portions of this segment are in different solvent environments. It is further proposed that this displacement may give rise to repacking of the HAMP. It is not clear, though, if the authors actually looked for any such repacking or not."

Following Reviewer's suggestion, we now analyzed the time evolution of helix curvature profile of TM2 residues in the AA and AH states of wild-type Tar and Tar^{TM2+2I} using MDAnalysis based on HELANAL algorithm (Fig. S11b, page 12, Refs. 78-79). In the AH state of Tar^{TM2+2I}, TM2 of the occupied monomer bends even further around the residue Leu205. The local angle of the residues shown in the helix curvature profile (Fig. S11b) is consistent with the bending angle of TM2 that we calculated before (Table S1, Fig. S11a). We have further added graphic illustration of signaling through TM2 and its inversion in Fig. 2d.

We also calculated the longitudinal distance of the residues in the control cable and AS1 to the lipid-cytoplasm interface (mass center of phosphorus atoms in cytoplasmic lipid leaflet) in the AA and AH states of wild-type Tar and Tar^{TM2+2I} (Fig. S12b-d, pages 12-13). The results are consistent with the longitudinal position of the residue Ile220 on AS1 relative to Ile220' on AS1' (Fig. S12a, Tables S2) and support our suggestions that the ligand-promoted asymmetric bending of TM2 of the occupied monomer in the AH state results in inward displacement of the control cable and AS1 in wild-type Tar, while leading to an outward displacement in Tar^{TM2+2I}.

Our more detailed analysis in the revised manuscript suggest that the protein-lipid electrostatic interactions and the sliding of the AS1 from the occupied monomer

influence the conformational dynamics of the HAMP domain (Fig. S6c, Fig. S14, Table S3, pages 14-15). We further compared the HAMP structures of the AA and AH states of wild-type Tar and Tar^{TM2+2I} and did not observe the helical rotation among the four helices.

"4) It's concerning that the authors report the breaking of the TM2 helix in the TM2+2I AH model near the middle of the membrane bilayer. Such a break requires significant energy in a properly equilibrated and tightly packed 4-helix transmembrane bundle. Did this happen in all of the simulations of this model? When in the simulations did the event take place? "

The restraints on the residues of the periplasmic domain in the AH states of wild-type Tar and Tar^{TM2+2I} were applied before the production run and maintained during the entire time of simulations to mimic binding of a ligand (Tables S8 and S10). It elicits the TM2 of Tar^{TM2+2I} bending even further around Leu205, the residue located near the end of TM2. The bending of TM2 breaks the helix structure around Leu205, which could be observed from beginning of simulations.

"5) Major issue: The whole of the conclusions drawn by the authors to explain the effects of the AS2/MH1 linker mutations for signaling inversion are dependent on RMSF measurements. While RMSF is a widely used pseudo-metric for describing protein fluctuations, it is heavily dependent on the particulars of the alignment and best suited for globular proteins. In particular, when dealing with long, coiled-coil bundles such as Tar, the natural, global bending of the bundle makes it difficult to align these structures in a way appropriate to allow for direct comparison of local RMSFs. No detailed description of the alignment process is given, rendering the present comparisons difficult to interpret. The lack of protocol detail here is especially concerning given the shape of the provided RMSF traces in Figure S12, which show nearly identical shapes differentiated by global shifts up or down. This implies that the changes in fluctuations are not internal to the domains themselves, but due to

some sort of global fluctuation possibly arising from misalignment. This severely undermines the offered piston-force/support-force explanation. A more convincing and interesting analysis would have involved the quantification of changes in the correlated motions between individual residues in the various regions of the HAMP/linker/MH bundle."

To rule out that our conclusions are due to the structure misalignment, we first conducted clustering of MD trajectories based on 3Å C α RMSD. By using the middle structure of the largest cluster as the reference, we calculated the RMSF of all the C α of each residue (Figs. S17 and S18). These new simulations are consistent with previous results.

Furthermore, as suggested by the Reviewer we analyzed the distance distributions of the residue pairs on the HAMP domain and the MH bundle in the AA and AH states of wild-type Tar and Tar^{HAMP_GVPQM} (Fig. S19, page 17). The residues on the position *f* of the *a-g* heptad of AS1, AS2, MH1 and MH2 were selected (Fig. S19a). The distance distributions of the residue pairs on the HAMP domain or MH bundle are also in line with the RMSF data (Fig. S19b-e).

"6) The simulations of the Tar periplasmic domain in two concentrations of NaCl appear to have been "thrown in" as an afterthought. The simulation analysis is not clearly described, nor is the robustness of the obtained results demonstrated. It is not clear how the two mentioned changes (inter-monomer rotation and sliding between helix a1 and a4) are actually "similar to the conformation triggered by the binding of aspartate," or what other changes are present in the structure. Though the experiments and logic used for mapping sensory regions in Tar are intriguing, considering the sparsity of MD-related detail, it's difficult to tell whether the presented simulations support the authors' conclusions or not."

We do believe that these simulations provide important insights into the mechanism of sodium and potassium sensing, suggesting that it is indeed similar to the conventional attractant response. To address Reviewer's comment, we now provided the C α RMSD plots for the three trajectories of the MD simulations for the Tar periplasmic domain at 0 and 50 mM NaCl (Fig. S23a). The systems remain stable with low RMSD values till the end of the simulations.

We also described the analysis procedures more clearly in the main text (pages 20-21). We performed backbone-RMSD clustering with a cutoff of 2.0 Å of all the conformations and compared the largest fractional cluster populations at 0 and 50 mM NaCl. The inward sliding of the α 4 helix relative to the α 1 helix and the inter-monomer rotation are the two dominant conformational changes in the receptor. They are in line with the two modes of conformational changes triggered by binding of aspartate, showing that binding of one attractant molecule to the Tar dimer induces both piston-like downward movements of the α 4 helix and relative rotations of the two monomers (Refs. 8-9), thus consistent with the attractant response of Tar to NaCl.

"In conclusion, the inverted signaling mutants presented are interesting experimentally as highlighted by the previous reviewers. In addition, the use of MD simulation in principal provides a solid framework to investigate the types of molecular questions proposed here, and the authors have proposed a rigorous set of MD simulations to investigate them. However, for the reasons cited above, the poor description of the equilibration process and analyses raise many questions that cannot be overlooked. More importantly, however, the authors miss numerous opportunities to provide new insight into transmembrane and HAMP signaling by largely limiting their analyses to those quantities already specified by Park et. al. (2011) and mechanistic concepts already proposed in the literature. As such, the MD results provide little new understanding or predictive power, and I unfortunately cannot recommend the article for publication in Nature Communications. "

As discussed above, we have now further verified and extended MD simulations, confirming our previous results and providing new insights into mechanisms of receptor signaling, particularly on the role of protein-lipid interactions and their effect on protein dynamics, differential effects of signaling on the dynamics of AS1 and AS2 of HAMP domain, roles of the ligand-occupied and ligand-free monomers of receptor, and the effects of signaling on the dynamics of methylation helices. We believe that this has substantially strengthened the modeling part of our work, and we thank the Reviewer for suggesting these additional analyses.

Reviewer #4 (replaces referee #2):

"My role as an additional reviewer was to comment on the revised manuscript and to assess the authors' response to reviewer #2 concerns.

Overall, this is an interesting paper with a large amount of experimental and simulation data. In my opinion, results do justify conclusions, especially after all the changes in response to reviewers' comments were made. Reviewers of the original submission did an excellent job in identifying potential issues and I have no specific concerns of my own.

Authors responded to reviewers' concerns adequately and additional data was provided where needed.

The main issue raised by Reviewer #2 was the impact of this study, which was viewed as modest. The authors, on the other hand, argued that their work provides mechanistic explanations for a phenomenon for the first time.

I agree with the authors that their work explains the mechanism of "the inverted response". Indeed, because of their work, this phenomenon can be understood in

molecular terms.

However, I also agree with Reviewer #2. Attempts to present this study as something bigger than that (e.g. plasticity, evolution) are not necessarily convincing. The fact that single mutations can dramatically alter protein function is well known, e.g. aberrant signaling by protein kinases leading to cancers.

To sum up, I do think that this paper will have impact on the chemotaxis field. Whether or not this impact is big enough to justify publication in Nature Communications is the Editor's call."

We thank the Reviewer for the generally positive evaluation of our work and for acknowledging its importance for the chemotaxis field. Concerning the evolutionary implications of our results, we would like to point out that already in the previous round of revision, the Discussion has been refocused on the insights into sensing and signaling, apart from one sentence in the first paragraph of the Discussion (“These results suggest that under selection bacterial sensors can easily change not only their ligand specificity but also the sign of their response.”) that directly restates our own observation in Fig. 3. The question how easy inversion of the response sign might happen is only raised in the Introduction and mentioned in the Abstract (which we now slightly modified to remove the word “evolutionary”), and we believe that raising this question is legitimate. Although the Reviewer is clearly right in saying that protein function can be dramatically altered by mutations, inversion of the signaling function is in our opinion much less trivial. And as stated in the Introduction, to our knowledge only very distantly related chemotactic systems of *E. coli* and *B. subtilis* are known to actually exhibit such inversion of receptor signaling. We thus do think that the ease with which such inverted mutants could be selected/obtained is surprising (and it is equally surprising that after so many years of chemotaxis research such mutations have not been characterized). But as mentioned, we are now already

careful not to overstate the evolutionary implications of our results and rather focus on mechanisms of signaling and sensing.

REVIEWERS' COMMENTS:

Reviewer #3 (Remarks to the Author):

The authors have largely addressed the requests for additional information pertaining to certain computational aspects of their work. Overall, I feel the manuscript has improved in these areas. However, considering the substantial overlap remaining with Park et al. (2011), especially in terms of the analyses employed and the general conclusions drawn, it does not seem that the authors have garnered much in the way of new understanding from their simulations. Ideally, refinements to the molecular mechanism presented in Park et al. would be suggested based on the observed trajectories or detailed residue-based predictions would be proposed for subsequent validating experiments. That being said, the simulation trends do appear consistent with the authors' experimental findings, and as such, they play a strong auxiliary role that strengthens the general narrative. Hence, the overall impact of the work, in my opinion, mostly boils down to the apparent significance of its experimental findings. These have already been discussed at some length by the other reviewers.

Point-by-point response to reviewers' comments

Reviewer #3 (Remarks to the Author):

"The authors have largely addressed the requests for additional information pertaining to certain computational aspects of their work. Overall, I feel the manuscript has improved in these areas. However, considering the substantial overlap remaining with Park et al. (2011), especially in terms of the analyses employed and the general conclusions drawn, it does not seem that the authors have garnered much in the way of new understanding from their simulations. Ideally, refinements to the molecular mechanism presented in Park et al. would be suggested based on the observed trajectories or detailed residue-based predictions would be proposed for subsequent validating experiments. That being said, the simulation trends do appear consistent with the authors' experimental findings, and as such, they play a strong auxiliary role that strengthens the general narrative. Hence, the overall impact of the work, in my opinion, mostly boils down to the apparent significance of its experimental findings. These have already been discussed at some length by the other reviewers."

We thank the Reviewer for acknowledging that we have largely addressed the requests for additional information regarding computational aspects of our work and that this has improved the manuscript. We also absolutely agree with the Reviewer that the main impact of our work is in the experimental findings and that the modelling plays an auxiliary role but further strengthens the manuscript. As the Reviewer also notices, the simulations are consistent with experimental findings, which further validates the presented model of signal transmission through chemoreceptors and enables us to identify and verify several important features of this model. Further expansion of the modelling part would go beyond the scope of our primarily experimental study.